

# Brief communication: Estimation of hydraulic properties of active layers using ground-penetrating radar (GPR) and 2D inverse hydrological modeling

Xicai Pan[1,2], Stefan Jaumann[1,3], Jiabao Zhang[2], and Kurt Roth[1,4]

[1]Institute of Environmental Physics, Heidelberg University, Im Neuenheimer Feld 229, 69120, Heidelberg, Germany
[2]Fengqiu Agro-ecological Experimental Station, State Key Laboratory of Soil and Sustainable Agriculture, Institute of Soil Science, Chinese Academy of Sciences, 71 East Beijing Road, Nanjing, China
[3]HGSMathComp, Heidelberg University, Im Neuenheimer Feld 205, 69120 Heidelberg, Germany
[4]Interdisplinary Center for Scientific Computing (IWR), Heidelberg University, Im Neuenheimer Feld 205, 69120 Heidelberg, Germany

*Correspondence to*: Xicai Pan (xicai.pan@issas.ac.cn)

**Abstract.** Estimation of hydraulic properties of active layers is challenging due to the freeze-thaw effect in space and time. Provided an active layer with an undulating frost table, monitoring of spatial soil water dynamics could provide significant information for commonly-used inverse estimation of soil hydraulic properties. In this study, we assess the feasibility of efficiently estimating soil hydraulic properties using Ground-Penetrating Radar (GPR) observations - soil water storage and layer thickness in conjunction with a two-dimensional hydrological model. The results of this study conceptually demonstrate that spatial and temporal observations of soil moisture in the active layer during a rain event are sufficient for inverse estimation of soil hydraulic parameters. The proposed method depends on the lateral water redistribution controlled by the undulating frost table and/or the intensity and duration of the precipitation. We suggest that this method could be used for seasonal-scale estimation of soil hydraulic properties.

## 1 Introduction

In permafrost regions, the active layer is a key stratum that controls the exchange of water and energy fluxes between the land surface and the atmosphere. However, the temporal development of the thawing front is hard to monitor at large scales because of its high spatial variability. The variability is not only a result of the local variation in surface features like microtopography and vegetation cover but is also related to subsurface soil properties which govern hydrological processes. Permafrost models are normally used to predict the evolution of active layer and permafrost thickness. However, their predictions for thawing depth and permafrost degradation are still barely satisfactory due to the low spatial resolution soil information like hydraulic properties and architecture (e.g., Pan et al., 2016). Therefore, gaining accurate soil properties is crucial for understanding permafrost degradation and the associated permafrost hydrology. This is particularly important for permafrost studies in the regions with thick and unsaturated active layers like the Qinghai-Tibetan Plateau (QTP). Soil



properties controlling active layer dynamics are the thermal and hydraulic capacities and conductivities. Since thermal properties are less variable than the hydraulic ones, they are normally estimated based on literature data. In contrast, the soil hydraulic properties are strongly site-specific. Thus, knowledge of the soil hydraulic properties is essential for permafrost modeling.

Modeling water dynamics in permafrost soils is difficult due to the highly nonlinear hydraulics. Particularly, the soil hydraulic properties are affected by considerable deformation and transport of soil material during freeze-thaw cycles (Ray et al., 1983; Boike et al., 1998). Additionally, lateral water redistribution is not negligible due to the high variability of the spatial thawing rate (e.g., Quinton et al., 2000; Wright et al., 2009). These features are adverse for the one-dimensional (1D) model, but this provides various possibilities for two-dimensional (2D) inverse modeling if spatial observations of the

thawing front and the soil moisture are available. These spatial observations contain significant information as they monitor soil water dynamics. Lateral water redistribution is common in thawing active layers with undulating frost tables, and happens continuously during the thawing season. This rapid lateral water redistribution also occurs after strong rainfall events, which is often seen in the permafrost regions on the QTP (Pan et al., 2014), where precipitation is dominated by summer monsoon.

Inverse methods are commonly used to estimate hydraulic properties from observed state variables at different scales. Progress in inverse modeling of soil hydraulic properties has been reviewed by Vrugt et al. (2008). Normally, the inverse method using in-situ 1D monitoring profile yields accurate data in depth, but it is expensive to apply to larger spatial scales. Conversely, recent developments of soil hydraulic property inversion using satellite remote sensing yield certain results at larger scales, e.g., from field-scale to catchment scale, which, however, are mostly based on the shallow measurement depth

(< 5 cm) and an assumption of homogeneous soil column (Mohanty, 2013). In fact, root zone soils are heterogeneous whereas observations of the top soils are not representative. Accordingly, the model predictions using estimated hydraulic parameters solely based on near-surface soil moisture observations are not as good as the ones based the observations for the entire profile (Bandara et al., 2014). In order to yield good results, all inverse parameter estimation methods rely on a time series of measurements that contain significant information on soil water dynamics (Bandara et al., 2013).

Geophysical methods like Ground-penetrating Radar (GPR) of detecting soil layer boundaries developed rapidly (e.g., Neal, 2004; Annan, 2005; Jol and Bristow, 2003) and was extended for in real-time imaging of near-surface/layering soil water content (e.g., Huisman et al., 2003; Weihermüller, et al., 2007; Bradford, 2008; Pan et al., 2012; Klenk et al., 2016). For permafrost soils, thawing depth of active layer and soil water content can be also simultaneously retrieved (e.g., Gerhards et al., 2008, Westermann et al., 2010). Quantitative observations of spatial-temporal variation in thawing depth and

soil water content within plot-scale soil were successfully demonstrated on the QTP by Wollschläger et al. (2010) and Pan et al. (2014). These methods could, in principle, supply the necessary data for inverse modeling of active layer water dynamics.

    Undulating frost tables are common in the permafrost regions resulting from the patterned surfaces such as vegetation cover, snow cover, and soil properties. Provided significant lateral water redistribution induced by an undulating frost table in active layers and spatial-temporal GPR observations, efficiently estimating effective hydraulic properties could be viable.





In this study, we investigate its feasibility using synthetic data, and discussed the controlling factors of this approach for practical applications.

## 2 Scheme of hydraulic parameter estimation

### 2.1 The 2D hydrological model

Generally, two-dimensional Darcian water flow in a variably-saturated isotropic medium is described with Richards equation (Richards, 1931)

$$\partial_t \theta(h) - \nabla \cdot \left[ K_w(\theta(h)) \left[ \nabla h - 1 \right] \right] = 0, \tag{1}$$

where $\theta$ is the volumetric soil water content, (m³m⁻³), $K_w$ is the hydraulic conductivity, (ms⁻¹), and $h$ is the matric head, (m). The material functions involving soil hydraulic parameters in the Richards equation compose the hydraulic conductivity

function $K_w$ (h) and the soil water characteristic $\theta$(h), usually in terms of the water saturation $\Theta$ (–). Widely employed models for these two relationships are the van Genuchten-Mualem models (van Genuchten, 1980; Mualem, 1976):

$$\Theta(h) = \frac{\theta(h) - \theta_r}{\theta_s - \theta_r} = (1 + |\alpha h|^n)^{-m} \tag{2}$$

$$K(\theta) = K_s \Theta^\tau [1 - (1 - \Theta^{1/m})^m]^2, \tag{3}$$

where $\theta_s$ and $\theta_r$ denote saturated and residual water contents, respectively, $\alpha$, $n$ and $m$ ($m = 1\text{-}1/n$) are empirical parameters

shaping the retention curve, $K_s$ is the hydraulic conductivity at saturation condition, and $\tau$ is an empirical parameter shaping the hydraulic function, which is commonly set as 0.5 (Mualem, 1976). With this, there are five unknown parameters $p = \{\alpha, \text{n}, \theta_r, \theta_s, K_s\}$.

The water dynamics in a structural active layer during a rainfall event can be simulated with following settings (Fig. 1). The model domain is defined as a rectangular grid with an undulating frost table. Materials are the same for the both layers but

the lower one is frozen. Hereby, their hydraulic properties, $K_s$ and $\alpha$, are approximated similar using a Miller scaling factor (Miller and Miller, 1956), and $n$, $\theta_r$ and $\theta_s$ are the same. A Neumann no-flow boundary condition is implemented at the bottom and both sides, and a Dirichlet condition is set for the upper boundary.

### 2.2 The parameter estimation

Given a time series of GPR observations of soil water content in the thawed active layer (Fig. 1c) and weather conditions,

the parameters were estimated with the following three steps.

In step 1, the van Genuchten parameters $p_1 = \{\alpha, \text{n}, \theta_r, \theta_s\}$ were estimated using the first GPR observation before the rainfall event, as inspired by the initial start of parameter estimation by Jaumann and Roth (2017). Since later water





redistribution leads to a higher water storage in the active layer with a deeper thawing depth, the 2D water storage distribution is correlated with the thawing structure. Thus, a static hydraulic equilibrium at the initial stage is assumed that a water table at the lower boundary. Correspondingly, the van Genuchten parameters can be derived by fitting the relationship between observed soil water content (storage) and matric potential (depth).

In step 2, the effective soil hydraulic parameters $p = \{\alpha, \mathrm{n}, \theta_\mathrm{r}, \theta_\mathrm{s}, K_\mathrm{s}\}$ were determined by minimizing the differences between observed water storages $l_\mathrm{obs}(x, t)$ and simulated water storages $l_\mathrm{mod}(x, p, t)$ at location x as an objective function

$$\chi^2(p) = \frac{1}{N}\frac{1}{M}\sum_{t}^{N}\sum_{x}^{M}\left[l_\mathrm{obs}(x, t) - l_\mathrm{mod}(x, p, t)\right]^2. \tag{4}$$

The Levenberg-Marquardt Algorithm as implemented in Jaumann and Roth (2017) is used to minimize $\chi^2(p)$. M: the number of the grids in x (M = 100), N: the number of observations (N = 7). Convergence typically requires less than 10 iterations.

In step 3, the final parameters $p_3 = \{\alpha, \mathrm{n}, \theta_\mathrm{r}, \theta_\mathrm{s}, K_\mathrm{s}\}$ were chosen from 50 ensemble inversions with a minimum $\chi^2$. As the Levenberg-Marquardt Algorithm is a gradient-based optimization method, it relies on good initial starting points of parameters. To address this, we used an ensemble of 50 samples for the initial parameter $K_\mathrm{s}$ in conjunction with the initial estimated van Genuchten parameters from step 1.

Overall, the framework of the 2D inversion procedure is shown in Fig. 2. In step 1, the parameter estimation was solved
using the function fminsearch from Matlab (Version: R2015a). In step 2, the Richards equation solver (muPhi, Ippisch et al., 2006) was used to simulate the spatiotemporal soil water dynamics, and the Levenberg-Marquardt algorithm was used to minimize the differences between the simulated state and observed state. In step 3, the final estimated parameters were determined from the 50 ensemble inversions.

## 3 Case studies

Here we used synthetic studies to verify the proposed approach, and assess the effects of undulating structure on its accuracy. Active layers (12 m x 2 m) composed a thawed layer and a frozen layer delineated by an undulating frost table (Fig. 3), and have the same sandy soil. The soil hydraulic parameters at thawed state are listed in Table 1. Following Miller scaling approach (Miller and Miller, 1956; Sposito and Jury, 1990; Roth, 1995), the parameters $n$, $\theta_\mathrm{r}$, and $\theta_\mathrm{s}$ for the frozen soil are the same as the thawed ones, but $K_\mathrm{s}$ and $\alpha$ are scaled with factors $10^{-2}$ and 10, respectively. The scaling factors are
assumed to be constant without considering their temperature dependency, since it is beyond the focus of this study.

Soil water dynamics in the domain was simulated with a time step of 100 minutes over a short period of 5.9 days. The time series of the upper forcing at the upper boundary by rainfall is shown in Fig. 1a. Seven snapshots of soil water storage observations were expressed by using the forward simulations and adding Gaussian distributed random errors of GPR observations. For a thawed layer (1 m) with a mean soil water content of 15%, we assume the uncertainty of depth
measurement is 0.05 m, and the uncertainty of water storage is deduced as 0.15×0.05 = 0.0075 m.





Since the non-uniform change of soil water storage is essential to the inversion, one controlling factor is the undulating structure of frost table. To investigate the influence of this, three active layers (S1, S2, and S3 in Fig. 3) with different undulating amplitudes, 0.25 m, 0.5 m, and 0.75 m, respectively, were investigated. In addition, to investigate the influence of the errors in GPR observations on the approach, parameter estimations were repeated 10 times using the GPR observations

with the same uncertainty but different random errors. All the inversions were conducted in a cluster via parallel computation.

## 4 Results and discussion

Figure 4 demonstrates the results of the parameter estimation. The left panel compares the estimated water retention curves of three structures (S1, S2, and S3) with the synthetic one after step 1. The middle and right panels compare the ensemble

estimates of water retention curve and hydraulic conductivity curve, respectively, with the synthetic ones after step 2. In each plot, the curves represent the best 34 estimates, namely, small $\chi^2$, accounting for 68.25% of the 50 ensembles, and the darker the curve, the better the estimation is. Results from three panels shows the order of the estimates for step 1 and step2 are S1 < S2 < S3. Generally, the proposed approach works well for the studied cases, and the larger undulating amplitude, the better the estimates. This is ascribed to the increasing intensity of lateral water redistribution.

The final estimates of parameters $p_3 = \left\{ \alpha,\ \mathrm{n},\ \theta_\mathrm{r},\ \theta_\mathrm{s},\ K_\mathrm{s} \right\}$ were derived from the best one, namely, smallest $\chi^2$, among the 50 ensembles through step 3. The effects of thawing structure on the inversion of the five van Genuchten-Mualem parameters are shown in Fig. 5. The histograms show the estimated parameters from step 3 with the ones from step 1. Generally, the improvement of step 1 to the van Genuchten parameters is quite limited, although the structure did exert impacts on the estimation. In contrast, the results from step 3 show the robustness of the method to the errors of GPR

observations, although the impacts of the structure are not negligible.

Overall, the proposed method works well for the studied cases, but it depends on the significance of lateral water redistribution, which is not only controlled by the structure of frost table but also precipitation features like intensity and duration. It is common that the inverse modeling of in-situ soil water dynamics under natural conditions relies on the wetting range, regulated by precipitation features (e.g., Steenpass et al., 2011; Scharnagl et al., 2011). Indeed, the larger the wetting

range, the better the constraint in the inversion.

For practical application, there are some necessary conditions for this approach to work. First of all, a continuous and undulating frost table is required that leads to lateral water redistribution. Applying this approach for a three-dimensional frost table could be handled in analogy, however. Secondly, for rain-based cases, prerequisite conditions like large rainfall intensity and good soil permeability are necessary, but they are only viable at specific regions, e.g., the northeastern QTP.

Finally, to capture the lateral water redistribution, selecting the time-slice observations is essential. As a rule of thumb, time-evenly distributed observations are best during the infiltration process. These requirements limit the range of application. Alternatively, the limitations can be alleviated when applying this approach at seasonal-scale, which, however, incorporates

evapotranspiration and variable frost table in the model. Since the process of lateral water redistribution does not rely only on precipitation but also melt water by soil thawing. The latter one is relatively slow, but the amount can be considerable in wet active layers. Here, the timing of GPR observations will not be as sensitive as in the rain-based application. We indeed expect this approach to work better at seasonal scale once the aforementioned improvements are of the 2D hydrological

model are implemented.

## 5 Summary

Frost tables are often undulating and cause a lateral redistribution of water within the active layer. We propose a method for the inverse estimation of soil hydraulic properties that exploits this situation. It employs GPR to measure the spatial distribution of liquid water and a 2D simulation of soil hydrology. This method is assessed using synthetic data. Based on a

single rain event, seven snapshots of soil water storage provided to be for a faithful inversion. The reasonable accuracy of the estimated parameters is as expected for the studied cases. This method depends on the magnitude of lateral water redistribution, which is controlled by the undulating frost table, by the soil hydraulic properties and by the intensity and duration of the precipitation.

As a conceptual study, we assume perfect model and observations. We comment that GPR works best in coarse-textured

soils and may not work at all silty or clay soils. This coincides with the applicability of the hydraulic method, however. Despite its limitations, this approach provides a useful means to efficiently estimate hydraulic parameters at the field scale. Its major advantages include non-destructive observations, a bigger scale of the soil hydraulic properties and efficiency in application.

## Acknowledgements

We acknowledge the support by the Deutsche Forschungsgemeinschaft (DFG): project RO 1080/12-2, and the Chinese Academy of Science funding project: 12201715143200200. The authors thank the support by the state of Baden-Württemberg through bwHPC and the German Research Foundation (DFG) through grant INST 35/1134-1 FUGG.

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





Table 1 Hydraulic parameters of a sand at the unfrozen condition and their allowed ranges for parameter estimation.

| symbol | description (unit) | value | allowed range |
|---|---|---|---|
| $\alpha$ | inverse of air entry suction (m$^{-1}$) | -2.0 | -10...-0.1 |
| n | measure of the pore-size distribution (-) | 4.0 | 1.3...8.0 |
| $\theta_r$ | residual water content (m$^3$ m$^{-3}$) | 0.0 | 0.001...0.02 |
| $\theta_s$ | saturated water content (m$^3$ m$^{-3}$) | 0.3 | 0.25...0.4 |
| $\log_{10}(K_s)$ | $K_s$: saturated hydraulic conductivity (m s$^{-1}$) | -4 | -6...-3 |





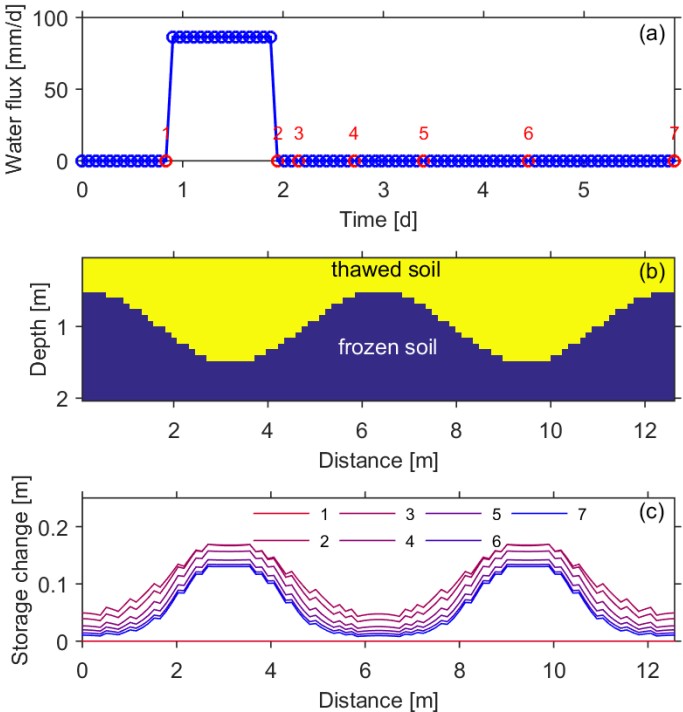

**Figure 1. An example of the forward modeling. (a) Water flux through the upper boundary. Red circles correspond to the snapshot timing of soil water storage observations in (c). (b) The structure of a thawing active layer. (c) Spatiotemporal variation of soil water storages in the thawed layer for different times.**



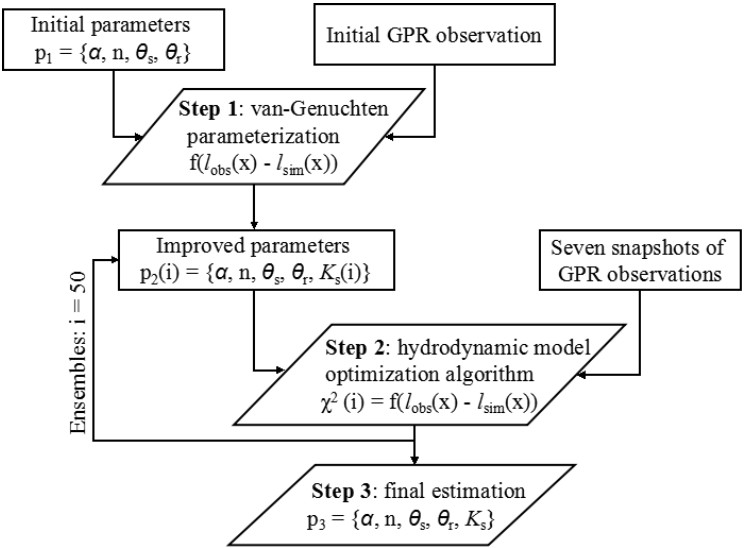

**Figure 2. The framework of the 2D inversion procedure.**





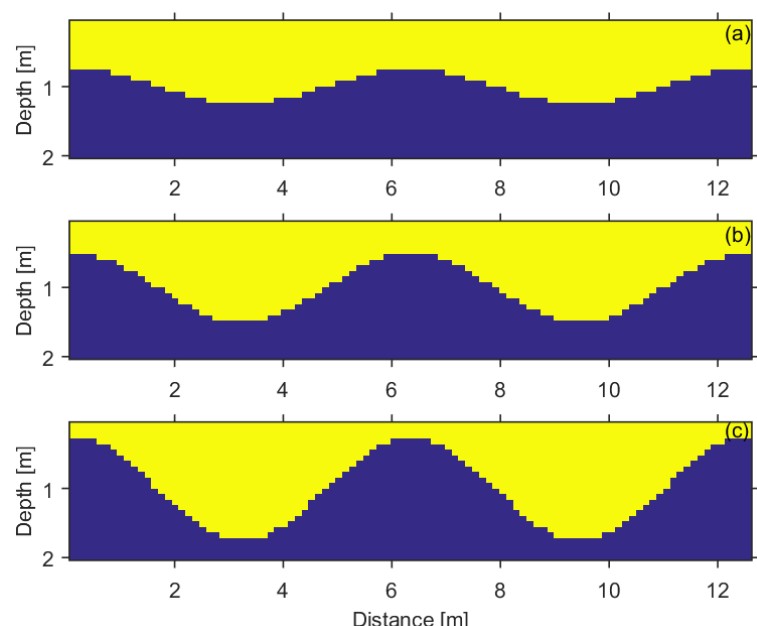

**Figure 3. The active layer with different thawing structures. The amplitudes of the undulating frost table are (a) S1: 0.25 m; (b) S2: 0.5 m, and (c) S3: 0.75 m, respectively.**



**Figure 4. A comparison of estimated water retention curves (step1 and step2) and hydraulic conductivity curve with the synthetic ones for three structures (S1, S2, and S3). Left panel: initial estimates of water retention curve. Middle panel: final estimates of water retention curve with the best 34 ensembles. Right panel: final estimates of hydraulic conductivity curve with the best 34 ensembles.**





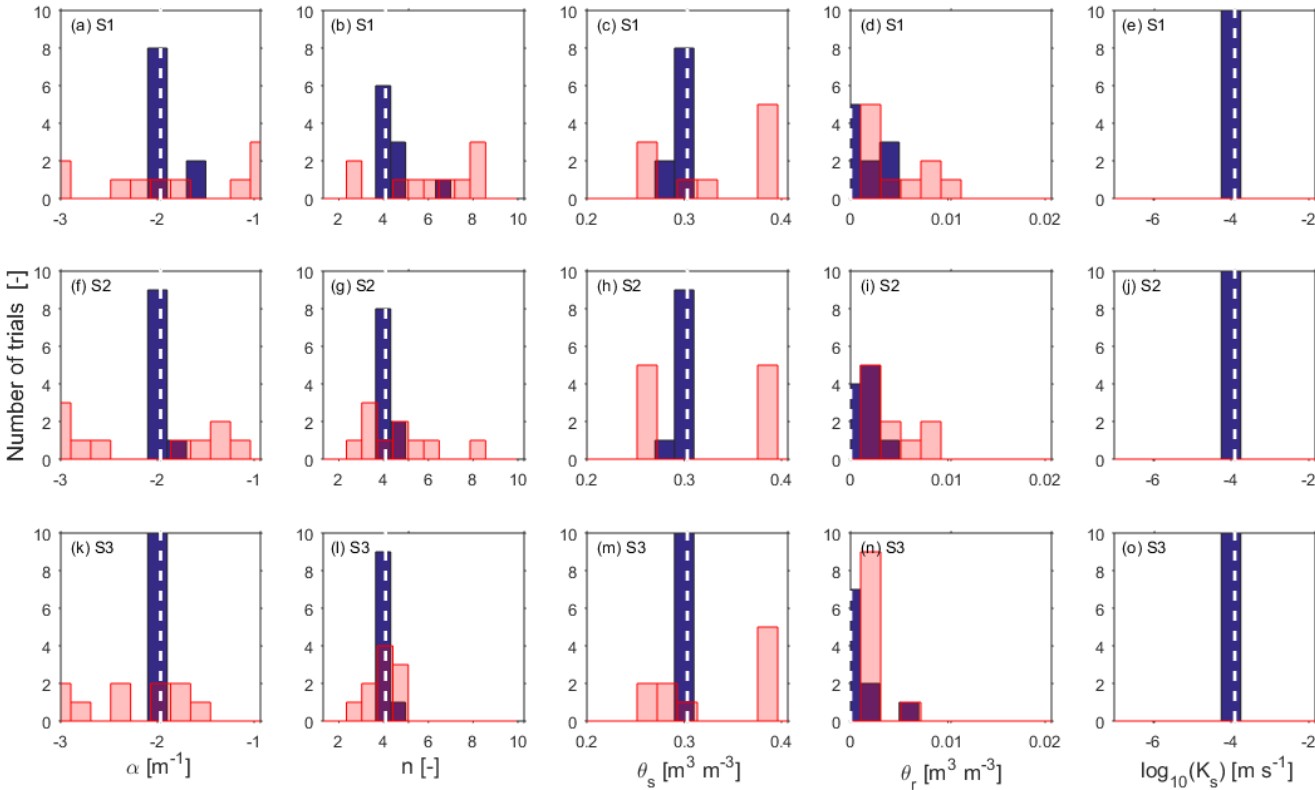

**Figure 5.** Effects of thawing structure on the inversion of the five van Genuchten-Mualem parameters. Given 10 repeated GPR observations with different random errors, histograms of the initial (step 1, light red bars) in and final (step 3, blue bars) estimated parameter values are shown. Only blue bars are shown in the right most panel because there are no initial estimates in step 1 for $K_s$. Dashed lines show the true parameter locations.