# Peer review of "Brief communication: Estimation of hydraulic properties of active layers using ground-penetrating radar (GPR) and 2D inverse hydrological modeling"

_The Cryosphere, 2017_

## Referee Comment (RC1) · Anonymous Referee #1 · 11 Jul 2017

**General Comments**

The authors analyze whether it is possible to identify soil hydraulic properties of the active layer in a permafrost region by inverse modeling using the Richards equation in two spatial dimensions. The content covers the scope of the journal, the study is conducted well in technical terms, and the manuscript is structured well and written in an acceptable style. From the point of view of inverse modeling in vadose zone hydrology, the study does thus not offer many new insights and the outcome is not surprising to me. An innovative feature is the investigation of the effect of the amplitude of the undulating frozen layer and its influence on parameter estimation. However, the

results of this are, again, overall not surprising. My main criticism can be summarized in two points:

1. The study uses only computer-generated data and assumes that the model is a perfect representation of the system. The impact of model error on the results is not investigated. Such model error could be caused by wrongly parameterized hydraulic properties or imperfect knowledge about boundary condition, initial conditions, and structural features of the soil. If the flow model is correct, the soil is homogeneous and data is only contaminated with independently, normally distributed noise of equal variance, the soil hydraulic properties can of course be identified under transient conditions and this is not worth reporting. In reality, these conditions will never be fulfilled in a field situation and the conclusion of the authors that their method can be applied for field data is thus not fully supported.

2. The study focuses only on the accuracy of the identified hydraulic properties, i.e. on the question how well the identified properties match the true ones. However, the aspect of precision or uncertainty is not treated well. I appreciate that the authors tested 10 different realizations of random error as stated on page 5 (top) and shown in Figure 5. Such a bootstrap is well-suited for quantifying uncertainties, but a bootstrap using only ten bootstrap samples cannot lead to a robust quantification of uncertainty.

I think it is absolutely necessary to analyze the influence of deviations from the almost perfect conditions assumed throughout the analysis and to improve the statistical quantification of uncertainties. Therefore, the authors should include the following aspects before publication:

1. Studies on the effect of more complex errors on the accuracy of the identified hydraulic properties (most importantly model error, but autocorrelated error is also an interesting aspect)

2. A more rigorous quantification of parameter uncertainty and parameter cross-correlation to delineate under which settings the unique identification of soil hydraulic properties of an active layer is possible.

3. A study using real GPR measurements to illustrate the performance of the proposed method in a real situation and to critically assess its potential and deficits.

**Specific Comments**

[General] An important point is whether the undulating structure of the frozen layer is also identified by the radar measurements or whether it is assumed to be known exactly, i.e. without error. In reality, it will be unknown and may deviate from the perfect shape assumed in this study. As a result, the soil hydraulic properties and the depths of the active layer as function of the horizontal variable must be identified jointly. This has not been investigated so far.

[P1 L18] "The proposed method depends on the lateral water distribution . . ." – what do you mean? In which sense does the method depend on it? Do you refer to applicability, accuracy, general results? Please be more precise.

[P2 L16] "Normally, the inverse method using in-situ 1D monitoring profile yields accurate data in depth, but it is expensive to apply to larger spatial scales." – what do you refer to exactly when you write "in-situ 1D monitoring profile"? Why is a 1D-method "expensive to apply at larger scales"? I don't understand what you mean, please clarify.

I do not understand why the hydraulic properties of the frozen layer are obtained by Miller-Miller scaling of the soil properties of the active layer (P3 L20; P4 L24). No justification is given for this. Why do you assume water flow in the frozen layer? I would assume that a frozen soil is impervious. Is it possible to describe

water flow in frozen soil with the Richards equation? Please mention the assumptions you make here and justify your approach.

[P3 L22] The authors use a Dirichlet condition at the top but do not mention the pressure head. I think that a flux boundary condition defined by the precipitation rate would be easier-to-implement and physically more realistic. Please justify the use of the Dirichlet condition and provide the pressure head value used in the simulations.

I miss information on the initial condition used in the numerical simulations (section 2.1). This is highly relevant for step 1 of the inverse procedure because the hydraulic properties are estimated using the assumption of a hydrostatic pressure distribution at the beginning. If a hydrostatic pressure head distribution was used as initial condition, step 1 becomes a trivial exercise, because the assumption of a hydrostatic pressure distribution made in step 1 is fulfilled. As a consequence, the results shown in the left three panels of figure 4 are not surprising.

Why do the authors use fminsearch for step 1 and Levenberg-Marquardt (LM) for step 2? I think LM is more efficient for step 1 than fminsearch which uses the Nelder-Mead-Simplex algorithm(NMS). The authors should mention the specific algorithm which fminsearch uses. The statement "As the Levenberg-Marquardt Algorithm is a gradient-based optimization method, it relies on good initial starting points of parameters." is misleading. The reason why LM needs good starting values is that it has only local convergence properties. The same holds for the NMS but this is not stated explicitly in the manuscript.

The authors state that they used "50 ensemble inversions" [P4 L10 L18]. I think the term ensemble is an exaggeration in this context. If I understand correctly what the authors did, they used different starting values for the model parameter Ks in the numerical minimization of the objective function and finally selected the one with the smallest value of the objective function. I would call this multistart LM minimization but not an ensemble inversion. The term ensemble is used in model

averaging or ensemble Kalman filtering but these techniques are much more so-phisticated compared to what the authors did. Neither do I understand the statis-tical background to show the best 34 functions in Figure 4. The optimization with the smallest value of the objective function is the maximum-likelihood-estimate and this is explicitly stated by the authors (P5 L15). But why would one include the next 33 results in the Figure? What is the statistical justification for this?

[P5 L 12] "Results from three panels shows the order of the estimates for step 1 and step2 are S1 < S2 < S3." – I do not understand what you mean with "S1 < S2 < S3". Do you mean that S3 is better than S2 than S1? How was this assessed? By the difference between the theoretical and identified hydraulic functions? If so, state it and provide some quantitative measure of goodness-of-fit, for instance root-mean-squared-error. I think such a statement on accuracy of the estimates must be complemented by a statement on the precision / un-certainty of the identified system properties. Such information can be based on the data shown in Figure 5, but the number of bootstrap samples is too small for statistical inference.

[Figure 5] Why are the results of the first step of the inverse method shown in Fig. 5? I thought that step 1 was used to obtain good initial estimates of the parameters for inversion steps 2 and 3. If this is correct, I don't see any reason to include the results of step 1 in Figure 5.

[P6 L12] "This method depends on the magnitude of lateral water redistribution, which is controlled by the undulating frost table, by the soil hydraulic properties and by the intensity and duration of the precipitation." – is this really a conclusion of your analysis? You have not varied rain intensity. Neither have you analyzed soil textures other than sandy. The only thing you have analyzed is the amplitude of the undulating frost table.

[Figure]

**Technical Corrections**

[P1 L13] "Provided an active layer with an undulating frost table, monitoring of spatial soil water dynamics" – incomplete sentence, please rephrase

[P1 L26] "Permafrost models" – consider to give a few examples and provide references or refer to a review article on such models.

[P1 L27] "due to the low spatial resolution soil information like hydraulic properties and architecture" – incomplete sentence, please rephrase. What do you mean exactly by architecture, structural features? Please rephrase.

[P2 L2] "they are normally estimated based on literature data" . I think the estimation is mostly based not only on literature data but additionally on texture information and empirical models. Please consider to rephrase.

[P2 L3] "Thus, knowledge of the soil hydraulic properties" – I don't think that this follows from the preceding sentence. Maybe you mean: "Thus, a site-specific determination of hydraulic properties is essential for permafrost modeling".

[P2 L18] "yield certain results" – of course they yield some results, but what do you mean? Do you mean "results of only limited accuracy" or results which are "only partly representative of the subsoil physical properties?" Please rephrase.

[P2 L23] "In order to yield good results, . . ." – the term "good" is not very specific, what do you mean, reliable, robust, accurate, . . .?

[P2 L29] "spatial-temporal" –> "spatiotemporal"

[P2 L33] "Provided significant lateral water redistribution induced by an undulating frost table in active layers and spatial-temporal GPR observations, efficiently estimating effective hydraulic properties could be viable" – is this sentence complete? Consider to rephrase.

[P3 Eq 1] The Richards equation is slightly wrong. The term "-1" is a scalar and thus cannot be added to the vector h in the square brackets.
[P3 L10] rephrase to "A widely applied model for these two relationships is the van Genuchten-Mualem model" (singular, not plural)

[P3 14] Provide units for the van Genuchten parameters in the text.

[P3 L20] "are approximated similar using" – please rephrase

[P4 L15] The authors provide a reference for the MuPhi solver by Ippisch but this reference is a bit misleading because the article by Ippisch et al. (2006, AWR) deals with a correction of van Genuchten-Mualem model close to saturation. Is it possible to give a more direct reference to the code? This would help the reader to access it.

[Fig 1] In the legend in the bottom plot (c), units are missing.

[P5 L18] "for rain-based cases" – please rephrase, it does not become clear what you mean.

[P6 L11] "The reasonable accuracy of the estimated parameters is as expected for the studied cases" – good that you have expected these results. But this is not a scientific statement. Would everyone expect them and if so: are they worth reporting? Please remove this statement and replace it.

---

## Referee Comment (RC2) · Anonymous Referee #2 · 7 Aug 2017

GENERAL COMMENTS

The manuscript describes a synthetic case study, in which a two-dimensional numerical model is used to estimate soil hydraulic parameters in idealized cross sections consisting of a permeable surface layer underlain by an impermeable bottom layer having undulating surface. Texts are generally well organized and written, and modelling approaches are technically sound. However, I am not convinced of the relevance and usefulness of the present manuscript in cryospheric science. It is an interesting numerical exercise, but the manuscript can be made much stronger and interesting to the readership of The Cryosphere. I will list my suggestions in specific comments

below.

SPECIFIC COMMENTS

Title. The study is motivated by GPR applications in the active layer, but the current version of the manuscript presents nothing specific to GPR or the active layer. For example, it does not deal with uncertainties and non-uniqueness in the relationship between dielectric permittivity, which is estimated by GPR, and volumetric water content. The model uses a two-layer structure consisting of permeable and impermeable soils separated by an undulating boundary, which is not specific to the active layer. The permeability of frozen soil is controlled by temperature, but the model does not account for coupled heat-energy transfer processes. Therefore, I think that the current title is somewhat misleading. It will be much better if the authors develop the paper to something that truly describes what is in the title.

P3, L7. This form of the Richards equation is incorrect. The gravity term should have a unit vector, not a scalar "1". Also, please define the direction of z-axis.

P3, L22. What value was used for the Dirichlet upper boundary condition? How was it determined?

P3, L25. GPR measures the travel time and amplitude of reflected radar waves, not the amount of soil water storage. The estimation of soil water storage from GPR data is not straight forward and has a large degree of uncertainty. To make this study relevant to The Cryosphere, it is highly desirable to incorporate uncertainties and non-uniqueness in GPR signal interpretation into numerical inversion. I believe that there is an established body of literature on this subject matter.

P3, L27 - P4, L2. I do not understand this sentence. Please rephrase.

P4, L2-3. It is assumed that the water table (i.e. matric potential = 0) is at the lower boundary. Does the lower boundary refer to the boundary between thawed and frozen soil? If so, does this "static condition" make sense hydrologically? For example, what

is the condition at time step 7 in Figure 1? Should that be a more logical representative of the static condition after the complete drainage of the active layer?

P4, L5. It appears that a homogeneous soil is used in the model. It is well known that the near surface soil in natural environments is highly heterogeneous both vertically and horizontally. This severely limits the usefulness of the proposed approach to determining soil hydraulic property. I see this as a major weakness of this manuscript. It can be made much stronger by explicitly treating soil heterogeneity in numerical inversion.

Figure 1. It appears that a constant flux was applied to the upper model boundary, whereas the method section states that the upper boundary had a Dirichlet condition (P3, L22). What was the actual boundary condition?

Table 1. The alpha values should be positive. The pore-size distribution coefficient (n) has a high value, and the residual water content is zero. I would say this is rather an unusual soil. Is this a good representative of typical soils in natural environments? Was this unusual soil purposely chosen for the synthetic case study? Why?

P5, L27-28. As the authors acknowledge, the water distribution over an irregular frost table is inherently three-dimensional. Two-dimensional models provide a useful tool for theoretical discussion, but its utility for practical application is limited. In addition, soil heterogeneity and a high degree of uncertainty in GPR data interpretation makes the present approach impractical to use in active-layer studies in natural environments. I suggest that the authors develop a full-length paper describing the development and application of a more realistic and useful inversion model using actual field examples of GPR data.

---

## Author Comment (AC1) · 12 Sep 2017

**Reply to Referee #1:**

**General comments**

*The authors analyze whether it is possible to identify soil hydraulic properties of the active layer in a permafrost region by inverse modeling using the Richards equation in two spatial dimensions. The content covers the scope of the journal, the study is conducted well in technical terms, and the manuscript is structured well and written in an acceptable style. From the point of view of inverse modeling in vadose zone hydrology, the study does thus not offer many new insights and the outcome is not surprising to me. An innovative feature is the investigation of the effect of the amplitude of the undulating frozen layer and its influence on parameter estimation. However, the results of this are, again, overall not surprising.*

**Reply:** We thank the reviewer for the constructive comments and suggestions. We revised the manuscript and thus refer to the revised manuscript.

We agree that the inverse modeling approach is not new in vadose zone hydrology and was applied already for various systems in the laboratory and also for some in the field. Building on previous works (e.g., Wollschläger et al., 2010; Pan et al., 2014) that repeatedly have shown that GPR can yield the stored water content and the layer depth simultaneously, we show that using these results together with the undulating subsurface architecture in the optimization procedure facilitates the accurate identification of effective soil hydraulic material properties. We agree that this idea is merely one step towards an efficient method to determine the effective soil hydraulic material properties of the active layer from GPR measurements. Hence, we changed the title and the manuscript accordingly. Although the results may not be surprising, this idea has - to the best of our knowledge - not been stated, used, or published somewhere else. Providing solutions for all the steps required for the complete endeavor involves considerable additional efforts and hence is beyond the scope of this brief communication.

*My main criticism can be summarized in two points:*

*1. The study uses only computer-generated data and assumes that the model is a perfect representation of the system. The impact of model error on the results is not investigated. Such model error could be caused by wrongly parameterized hydraulic properties or imperfect knowledge about boundary condition, initial conditions, and structural features of the soil. If the flow model is correct, the soil is homogeneous and data is only contaminated with independently, normally distributed noise of equal variance, the soil hydraulic properties can of course be identified under transient conditions and this is not worth reporting. In reality, these conditions will never be fulfilled in a field situation and the conclusion of the authors that their method can be applied for field data is thus not fully supported.*

**Reply:** This work assumes that GPR measurements can provide the stored water content and the layer depth simultaneously. It has been shown in previous works that this is possible. Typically, these results yield properties for effective homogenous subsurface materials. We agree that if the heterogeneities of the subsurface are large, the presented evaluation method is not valid. Fortunately, the evaluation of the GPR radargrams also reveals the degree of heterogeneity of the subsurface. According, e.g., to Wollschläger et al. (2010) and Pan et al. (2014), the heterogeneities typically are small and hence effective homogeneous material properties suffice for a quantitative understanding of soil water movement on larger scales. As we use the integrated water storage as input data for the optimization, we assume in this work that the remaining model errors can be represented with a white Gaussian noise. We agree that this does not necessarily need to be the case for measured data. However, this would then be identifiable analyzing the residuals of the inversion. Such an analysis would give rise to improvements of the representation (e.g., along the lines of Jaumann and Roth (2017)) or even to the use of Data Assimilation approaches, e.g., including the estimation of the boundary condition similar to Bauser et al. (2016). Such more powerful and expensive methods are used whenever demanded by the data. Here, they are beyond scope.

*2. The study focuses only on the accuracy of the identified hydraulic properties, i.e. on the question how well the identified properties match the true ones. However, the aspect of precision or uncertainty is not treated well. I appreciate that the authors tested 10 different realizations of random error as stated on page 5 (top) and shown in Figure 5. Such a bootstrap is well-suited for quantifying uncertainties, but a bootstrap using only ten bootstrap samples cannot lead to a robust quantification of uncertainty.*

**Reply:** Please note that in this study we do not only analyze the accuracy of the identified hydraulic properties but also the effect of the amplitude of the undulating frozen layer being a necessary condition for application of the proposed approach. We initially also ran the study using 5 different realizations of the

observation errors yielding qualitatively the same results (not shown, however). Therefore, we think 10 different realizations capture the main effect of the random error on the accuracy of the identified hydraulic properties. The current study comprises 50 ensemble members, 3 subsurface architectures, and 10 different realizations of observation errors making it a total of 1500 2D Levenberg-Marquardt inversion runs estimating 5 parameters. This required running 192 processors for 10 days.

*I think it is absolutely necessary to analyze the influence of deviations from the almost perfect conditions assumed throughout the analysis and to improve the statistical quantification of uncertainties. Therefore, the authors should include the following aspects before publication:*
*1. Studies on the effect of more complex errors on the accuracy of the identified hydraulic properties (most importantly model error, but autocorrelated error is also an interesting aspect)*
**Reply:** As mentioned in the reply to the point 1 above, we assume that we consider the most important model errors when using GPR observations which are according to the GPR measurement data (e.g., Wollschläger et al., 2010 and Pan et al., 2014). In cases where this assumption is not valid, the parameter estimation procedure would reveal representation errors through the residuals. The appropriate steps to improve the represenation in order to decrease these residuals typically depend on the specific setup of the experiment or the applied parameter estimation procedure. A complete investigation of all possible model errors could be useful but is out of the scope of this brief communication.

The autocorrelated error among hydraulic parameters has been investigated in our algorithm, but we omitted relevant description in the original manuscript. We add it in the revised manuscript (See next reply).

*2. A more rigorous quantification of parameter uncertainty and parameter crosscorrelation to delineate under which settings the unique identification of soil hydraulic properties of an active layer is possible.*
**Reply:** We improved the analysis of the parameter uncertainty in P5 L23-32 and P6 L1-L13 and also show the parameter correlations of the best ensemble member in Table 2.

*3. A study using real GPR measurements to illustrate the performance of the proposed method in a real situation and to critically assess its potential and deficits*
**Reply:** Estimating soil hydraulic material properties of the active layer based on GPR measurements is the final goal connected with this work. We agree that we here merely present one step towards this goal and changed the title accordingly. However, it has been shown that the assumptions on the data used in the presented method can be satisfied (e.g., Wollschläger et al., 2010 and Pan et al., 2014).

**Specific comments**

*[General] An important point is whether the undulating structure of the frozen layer is also identified by the radar measurements or whether it is assumed to be known exactly, i.e. without error. In reality, it will be unknown and may deviate from the perfect shape assumed in this study. As a result, the soil hydraulic properties and the depths of the active layer as function of the horizontal variable must be identified jointly. This has not been investigated so far.*
**Reply:** Available GPR evaluation methods simultaneously yield spatiotemporal water dynamics and the undulating structure of the frozen layer (e.g., Wollschläger et al., 2010 and Pan et al., 2014). Hence, it is not necessary to identify the subsurface architecture and the material properties jointly. Typically, the relative error for the depth is usually less than 5%, while the relative error for soil water content is much higher, e.g., 5-20%. Thus, we assume in this study that the undulating structure is known exactly and assign the associated error to the error of the soil water storage measurements.

*[P1 L18] "The proposed method depends on the lateral water distribution ..." – what do you mean? In which sense does the method depend on it? Do you refer to applicability, accuracy, general results? Please be more precise.*
**Reply:** The sentence was rephrased as "The applicability of the proposed method depends on the lateral water distribution"

*[P2 L16] "Normally, the inverse method using in-situ 1D monitoring profile yields accurate data in depth, but it is expensive to apply to larger spatial scales." – what do you refer to exactly when you write "in-situ*

*1D monitoring profile"? Why is a 1D-method "expensive to apply at larger scales"? I don't understand what you mean, please clarify.*

**Reply:** The sentence was rephrased as "Normally, the inverse method using in-situ 1D monitoring profile yields accurate data in depth, but its spatial representativeness is very limited."

*I do not understand why the hydraulic properties of the frozen layer are obtained by Miller-Miller scaling of the soil properties of the active layer (P3 L20; P4 L24).*

**Reply:** Here the Miller-Miller scaling is used due to numerical reasons. The sentence is rephrased in P4 L28-30 as "Due to numerical reasons, the according hydraulic conductivity has to be larger than 0. Hence we chose to scale the parameters of the active layer with a Miller scaling factor of 10 ensuring a constant hydraulic conductivity which is 1% of the one of the thawed active layer."

*No justification is given for this. Why do you assume water flow in the frozen layer? I would assume that a frozen soil is impervious. Is it possible to describe water flow in frozen soil with the Richards equation? Please mention the assumptions you make here and justify your approach.*

**Reply:** We agree with that assuming an impermeable frozen soil would better. However, limited by the capability of our model, we set an unfrozen layer with extremely low permeability to be equivalent to the frozen layer. Accordingly, the use of the Richards equation is suitable in the modeling. We think this is reasonable.

*[P3 L22] The authors use a Dirichlet condition at the top but do not mention the pressure head. I think that a flux boundary condition defined by the precipitation rate would be easier-to-implement and physically more realistic. Please justify the use of the Dirichlet condition and provide the pressure head value used in the simulations.*

**Reply:** Thanks for pointing out the mistake. We did use a flux boundary condition. The boundary conditions were rephrased in P4 L30 and P5 L1-2 as "The infiltration is represented with a Neumann flux at the upper boundary and the groundwater table at the lower boundary is represented setting the Dirichlet potential to 0. The boundaries at the sides are impermeable."

*I miss information on the initial condition used in the numerical simulations (section 2.1). This is highly relevant for step 1 of the inverse procedure because the hydraulic properties are estimated using the assumption of a hydrostatic pressure distribution at the beginning. If a hydrostatic pressure head distribution was used as initial condition, step 1 becomes a trivial exercise, because the assumption of a hydrostatic pressure distribution made in step 1 is fulfilled. As a consequence, the results shown in the left three panels of figure 4 are not surprising.*

**Reply:** Concerning the actual application of GPR measurements for our purpose, the initial measurement is usually conducted in a nearly equilibrium condition in permafrost regions. Thus it is fair well to assume a hydrostatic pressure head distribution as an initial condition.

*Why do the authors use fminsearch for step 1 and Levenberg-Marquardt (LM) for step 2? I think LM is more efficient for step 1 than fminsearch which uses the Nelder-Mead-Simplex algorithm(NMS). The authors should mention the specific algorithm which fminsearch uses. The statement "As the Levenberg-Marquardt Algorithm is a gradient-based optimization method, it relies on good initial starting points of parameters." is misleading. The reason why LM needs good starting values is that it has only local convergence properties. The same holds for the NMS but this is not stated explicitly in the manuscript.*

**Reply:** The reason for using two different algorithms is that we run step 1 and all post processing in the Matlab environment, while the inverse modeling was conducted in a separate package using C++ code. We agree that both algorithms rely on good starting values. However, their convergence also depends on the problem at hand.

In step 1, we just fit the retention curve with observations of water content and matric head. Typically, this is a rather well-posed and convex problem and we do not expect local minima in the investigated parameter space. As the forward model corresponds to a simple evaluation of a function, the easy application of fminsearch in the Matlab environment compensates for losses in performance.

In step 2, however, the problem is typically not well-posed and we expect local minima in the investigated parameter space. Hence, we choose an ensemble approach and sample the initial parameters with the Latin hypercube algorithm. Additionally, the forward model requires the solution of the Richards

equation in 2D, therefore we choose the Levenberg-Marquardt algorithm here.

*The authors state that they used "50 ensemble inversions" [P4 L10 L18]. I think the term ensemble is an exaggeration in this context. If I understand correctly what the authors did, they used different starting values for the model parameter Ks in the numerical minimization of the objective function and finally selected the one with the smallest value of the objective function. I would call this multistart LM minimization but not an ensemble inversion. The term ensemble is used in model averaging or ensemble Kalman filtering but these techniques are much more sophisticated compared to what the authors did. Neither do I understand the statistical background to show the best 34 functions in Figure 4. The optimization with the smallest value of the objective function is the maximum-likelihood-estimate and this is explicitly stated by the authors (P5 L15). But why would one include the next 33 results in the Figure? What is the statistical justification for this?*

**Reply:** We first remind that we use different starting values for all the model parameters $p = \{\alpha,\ n,\ \theta_r,\ \theta_s,\ K_s\}$ in the numerical minimization of the objective function, and the finial values are selected from the one with the smallest value of the objective function among the 50 ensembles. Regarding the term ensemble, we cannot agree with the reviewer. Ensemble means a group producing a single effect (https://www.merriam-webster.com/dictionary/ensemble).

As stated in P4 L18-19, we select the best 34 estimates, namely, small $\chi^2$, accounting for 68% of the 50 ensemble inversions in Fig. 4, i.e., 1-sigma of a Gaussian distribution. This depicts the goodness-of-fit of the 50 inversions by excluding some outliers caused by the local convergence of the LM algorithm.

*[P5 L 12] "Results from three panels shows the order of the estimates for step 1 and step2 are S1 < S2 < S3." – I do not understand what you mean with "S1 < S2 < S3". Do you mean that S3 is better than S2 than S1? How was this assessed? By the difference between the theoretical and identified hydraulic functions? If so, state it and provide some quantitative measure of goodness-of-fit, for instance root-mean-squared-error. I think such a statement on accuracy of the estimates must be complemented by a statement on the precision / uncertainty of the identified system properties. Such information can be based on the data shown in Figure 5, but the number of bootstrap samples is too small for statistical inference.*

**Reply:** Here the order stands for goodness-of-fit, which is based on a quantitative measure of the minimum cost function value. Considering the feature of local convergence of the L-M algorithm, we just choose the ensemble inversions with cost function values within 1-sigma of all (34 in 50) by excluding outliers. Figure 4 just provides an example of the results from step 1 and step 2 (only show 34 ensemble inversions). It mainly shows the effects of initial starts of parameters on the results. Whereas, Figure 5 shows the effects of GPR observations with random errors on the results from step1, step 2 and step3, in which only the best ensemble inversion for step 2 is used. Therefore, we use the standard error of all ensembles (10) as the measure of the uncertainty of the approach. Relevant statements are rephrased in P4 L18-19 and P5 L28-32.

*[Figure 5] Why are the results of the first step of the inverse method shown in Fig. 5? I thought that step 1 was used to obtain good initial estimates of the parameters for inversion steps 2 and 3. If this is correct, I don't see any reason to include the results of step 1 in Figure 5.*

**Reply:** Our aim of Figure 5 is to show the improvements of parameter estimation from Step 1 to step 3. Step 1 just provides a rough estimate of the parameters, and they are used as starting points of parameters for inversion step 2. Step 3 yields the final estimates derived from the 50 ensemble inversions. Overall, Figure 5 clearly shows the impact of the uncertainty in GPR observations in three steps.

*[P6 L12] "This method depends on the magnitude of lateral water redistribution, which is controlled by the undulating frost table, by the soil hydraulic properties and by the intensity and duration of the precipitation." – is this really a conclusion of your analysis? You have not varied rain intensity. Neither have you analyzed soil textures other than sandy. The only thing you have analyzed is the amplitude of the undulating frost table.*

**Reply:** We agree and the sentence is rephrased as "The applicability of this method depends on the site of lateral water redistribution, which is mainly controlled by the undulating frost table in the studied case."

**Technical Corrections**

*[P1 L13] "Provided an active layer with an undulating frost table, monitoring of spatial soil water dynamics" – incomplete sentence, please rephrase*

**Reply:** We agree. The sentence is rephrased as "Provided an active layer with an undulating frost table, monitoring of spatial soil water dynamics in the thawed layer could provide significant information for commonly-used inverse estimation of soil hydraulic properties.".

*[P1 L26] "Permafrost models" – consider to give a few examples and provide references or refer to a review article on such models.*

**Reply:** We agree. The sentence is rephrased as "Permafrost models such as empirical and process-based ones are normally used to predict the evolution of active layer and permafrost thickness (Riseborough et al., 2008).".

Riseborough, D., Shiklomanov, N., Etzelmuller, B., Gruber, S., and Marchenko, S.: Recent advances in permafrost modeling, Permafrost Periglac., 19 (2), 137-156, 2008.

*[P1 L27] "due to the low spatial resolution soil information like hydraulic properties and architecture" – incomplete sentence, please rephrase. What do you mean exactly by architecture, structural features? Please rephrase.*

**Reply:** We agree. The sentence is rephrased as "However, their predictions for thawing depth and permafrost degradation are still barely satisfactory, as this, e.g., lack of high spatial resolution of soil information like hydraulic properties."

*[P2 L2] "they are normally estimated based on literature data" . I think the estimation is mostly based not only on literature data but additionally on texture information and empirical models. Please consider to rephrase.*

**Reply:** We agree. The sentence is rephrased as "Since thermal properties are less variable than the hydraulic ones, they are mostly derived from soil texture information and empirical models as well as literature data.".

*[P2 L3] "Thus, knowledge of the soil hydraulic properties" – I don't think that this follows from the preceding sentence. Maybe you mean: "Thus, a site-specific determination of hydraulic properties is essential for permafrost modeling".*

**Reply:** We agree. The two sentences are rephrased as "In contrast, the soil hydraulic properties are very sensitive to small variation in soil texture or even small-scale structural patterns. Thus, a site-specific determination of hydraulic properties is essential for permafrost modeling.".

*[P2 L18] "yield certain results" – of course they yield some results, but what do you mean? Do you mean "results of only limited accuracy" or results which are "only partly representative of the subsoil physical properties?" Please rephrase.*

**Reply:** We agree. The sentence is rephrased as "Generally, results from the inverse method using in-situ 1D monitoring profile are limited to point scale, and this approach is usually hard to used larger scales."

*[P2 L23] "In order to yield good results, ..." – the term "good" is not very specific, what do you mean, reliable, robust, accurate, ...?*

**Reply:** We agree. The sentence is rephrased as "Furthermore, to yield robust results, all inverse parameter estimation methods rely on a time series of measurements that must contain significant information on soil water dynamics than (Bandara et al., 2013).".

*[P2 L29] "spatial-temporal" –> "spatiotemporal"*

**Reply:** We changed the manuscript accordingly.

*[P2 L33] "Provided significant lateral water redistribution induced by an undulating frost table in active layers and spatial-temporal GPR observations, efficiently estimating effective hydraulic properties could be viable" – is this sentence complete? Consider to rephrase.*

**Reply:** We agree. The sentence is rephrased as "Provided significant lateral water redistribution induced by

an undulating frost table in active layers and spatiotemporal GPR observations, we investigate the applicability of the inverse modeling approach to efficiently estimate effective hydraulic properties."

*[P3 Eq 1] The Richards equation is slightly wrong. The term "-1" is a scalar and thus cannot be added to the vector h in the square brackets.*
**Reply:** We changed the manuscript accordingly.

*[P3 L10] rephrase to "A widely applied model for these two relationships is the van Genuchten-Mualem model" (singular, not plural)*
**Reply:** We changed the manuscript accordingly.

*[P3 14] Provide units for the van Genuchten parameters in the text.*
**Reply:** We changed the manuscript accordingly.

*[P3 L20] "are approximated similar using" – please rephrase*
**Reply:** The sentence is removed. See more reply for Specific Commons 4.

*[P4 L15] The authors provide a reference for the MuPhi solver by Ippisch but this reference is a bit misleading because the article by Ippisch et al. (2006, AWR) deals with a correction of van Genuchten-Mualem model close to saturation. Is it possible to give a more direct reference to the code? This would help the reader to access it.*
**Reply:** The reference was chosen upon the advice of Olaf Ippisch. The article has a short paragraph that explains which methods are used in the implementation of muPhi. We received the software via personal communication.

*[Fig 1] In the legend in the bottom plot (c), units are missing.*
**Reply:** We agree. The caption is rephrased as "Figure 1. An example of the forward modeling. (a) Water flux through the upper boundary. Red circles show the times of seven snapshots of soil water storage observations. (b) The structure of a thawing active layer. (c) A time series of soil water storage observations in the thawed layer, corresponding to the time markers in (a)."

*[P5 L18] "for rain-based cases" – please rephrase, it does not become clear what you mean.*
**Reply:** We agree. It is rephrased as "for the studied case".

*[P6 L11] "The reasonable accuracy of the estimated parameters is as expected for the studied cases" – good that you have expected these results. But this is not a scientific statement. Would everyone expect them and if so: are they worth reporting? Please remove this statement and replace it*
**Reply:** We agree. It is rephrased as "The reasonable accuracy of the estimated parameters indicates the good feasibility of the proposed method at desired conditions."

**Brief communication: Toward the estimation of hydraulic properties of active layers using ground-penetrating radar (GPR) and inverse hydrological modeling**

Xicai Pan[1,2], Stefan Jaumann[1,3], Jiabao Zhang[2], and Kurt Roth[1,4]

[1]Institute of Environmental Physics, Heidelberg University, Im Neuenheimer Feld 229, 69120, Heidelberg, Germany
[2]Fengqiu Agro-ecological Experimental Station, State Key Laboratory of Soil and Sustainable Agriculture, Institute of Soil Science, Chinese Academy of Sciences, 71 East Beijing Road, Nanjing, China
[3]HGSMathComp, Heidelberg University, Im Neuenheimer Feld 205, 69120 Heidelberg, Germany
[4]Interdisplinary Center for Scientific Computing (IWR), Heidelberg University, Im Neuenheimer Feld 205, 69120 Heidelberg, Germany

*Correspondence to*: Xicai Pan (xicai.pan@issas.ac.cn)

**Abstract.** Estimation of hydraulic properties of active layers is challenging due to the freeze-thaw effect in space and time. Provided an active layer with an undulating frost table, monitoring of spatial soil water dynamics in the thawed layer could provide significant information for commonly-used inverse estimation of soil hydraulic material properties. In this study, we assess the feasibility to estimate effective soil hydraulic properties using two-dimensional measurements of soil water storage and thawing depth, which can be efficiently derived from ground-penetrating radar (GPR) observations. The results of this study conceptually demonstrate that spatial and temporal observations of soil moisture in the active layer during a rain event are sufficient for inverse estimation of soil hydraulic parameters. The proposed method depends on the lateral water redistribution controlled by the undulating frost table. We suggest that this method could be used for seasonal-scale estimation of soil hydraulic properties.

**1 Introduction**

In permafrost regions, the active layer is a key stratum that controls the exchange of water and energy fluxes between the land surface and the atmosphere. However, the temporal development of the thawing front is hard to monitor at large scales because of its high spatial variability. The variability is not only a result of the local variation in surface features like microtopography and vegetation cover but is also related to subsurface soil properties which govern hydrological processes. Permafrost models such as empirical and process-based ones are normally used to predict the evolution of active layer and permafrost thickness (Riseborough et al., 2008). However, their predictions for thawing depth and permafrost degradation are still barely satisfactory, as this, e.g., lack of high spatial resolution of soil information like hydraulic material properties. Therefore, gaining accurate soil properties is crucial for understanding permafrost degradation and the associated permafrost hydrology. This is particularly important for permafrost studies in the regions with thick and unsaturated active layers like the Qinghai-Tibetan

Plateau (QTP). Soil properties controlling active layer dynamics are the thermal and hydraulic capacities and conductivities. Since thermal properties are less variable than the hydraulic ones, they are mostly derived from soil texture information and empirical models as well as literature data. In contrast, the soil hydraulic material properties are very sensitive to small variation in soil texture or even small-scale structural patterns. Thus, a site-specific determination of hydraulic material properties is
5    essential for permafrost modeling.

Modeling water dynamics in permafrost soils is difficult due to the highly nonlinear hydraulics. Particularly, the soil hydraulic material properties are affected by considerable deformation and transport of soil material during freeze-thaw cycles (Ray et al., 1983; Boike et al., 1998). Additionally, lateral water redistribution is not negligible due to the high variability of the spatial thawing rate (e.g., Quinton et al., 2000; Wright et al., 2009). These features are adverse for the one-dimensional
10   (1D) model, but this provides various possibilities for two-dimensional (2D) inverse modeling if spatial observations of the thawing front and the soil moisture are available. These spatial observations contain significant information as they monitor soil water dynamics. Lateral water redistribution is common in thawing active layers with undulating frost tables, and happens continuously during the thawing season. This rapid lateral water redistribution also occurs after strong rainfall events, which is often seen in the permafrost regions on the QTP (Pan et al., 2014), where precipitation is dominated by summer monsoon.
15   Inverse methods are commonly used to estimate hydraulic material properties from observed state variables at different scales. Progress in inverse modeling of soil hydraulic properties has been reviewed by Vrugt et al. (2008). Generally, results from the inverse method using in-situ 1D monitoring profile are limited to point scale, and this approach is usually hard to be used larger scales. Conversely, recent developments of soil hydraulic property inversion using satellite remote sensing yield some useful results at larger scales, e.g., from field-scale to catchment scale, which, however, are mostly based on the shallow
20   measurement depth (< 5 cm) and an assumption of homogeneous soil column (Mohanty, 2013). In fact, root zone soils are heterogeneous whereas observations of the top soils are not representative. Accordingly, the model predictions using estimated hydraulic parameters solely based on near-surface soil moisture observations are not as good as the ones based the observations for the entire profile (Bandara et al., 2014). Furthermore, to yield robust results, all inverse parameter estimation methods rely on a time series of measurements that must contain significant information on soil water dynamics (Bandara et al., 2013).
25   Geophysical methods like ground-penetrating radar (GPR) of detecting soil layer boundaries developed rapidly (e.g., Neal, 2004; Annan, 2005; Jol and Bristow, 2003) and was extended for in real-time imaging of near-surface/layering soil water content (e.g., Huisman et al., 2003; Weihermüller, et al., 2007; Bradford, 2008; Pan et al., 2012b; Klenk et al., 2016). For permafrost soils, thawing depth of active layer and soil water content can be also simultaneously retrieved (e.g., Gerhards et al., 2008, Westermann et al., 2010), hence the total soil water storage can be derived accordingly as an integrated value of
30   both. Quantitative observations of spatiotemporal variation in thawing depth and soil water content within plot-scale soil were successfully demonstrated on the QTP by Wollschläger et al. (2010) and Pan et al. (2014). As a primary modeling step, we go for a representation of the possibly heterogeneous soil as a set of uniform layers. The object then is to determine the effective hydraulic material properties, which we here propose to do based on GPR measurements. Whether such an effective representation is reasonable or not can be judged from the measured radargram.

Undulating frost tables are common in the permafrost regions resulting from the patterned surfaces such as vegetation cover, snow cover, and soil properties. Provided significant lateral water redistribution induced by an undulating frost table in active layers and spatiotemporal GPR observations, we investigate the applicability of the inverse modeling approach to efficiently estimate effective hydraulic material properties. In this study, we use synthetic data and discuss the effect of the amplitude of the undulating frost table on the estimating effective soil hydraulic material properties.

**2 Scheme of hydraulic parameter estimation**

**2.1 The 2D hydrological model**

Generally, two-dimensional Darcian water flow in a variably-saturated isotropic medium is described with Richards equation (Richards, 1931)

$$\partial_t \theta(h) - \nabla \cdot \left[ K_w(\theta(h)) \left[ \nabla h - e_z \right] \right] = 0, \tag{1}$$

where $\theta$ is the volumetric soil water content, ($m^3 m^{-3}$), $K_w$ is the hydraulic conductivity, ($ms^{-1}$), and $h$ is the matric head, (m), unit vector in z-direction $e_z$ indicating the direction of gravity. The material functions involving soil hydraulic parameters in the Richards equation compose the hydraulic conductivity function $K_w$ (h) and the soil water characteristic $\theta$(h), usually in terms of the water saturation $\Theta$ (–). A widely employed model for these two relationships is the van Genuchten-Mualem model (van Genuchten, 1980; Mualem, 1976):

$$\Theta(h) = \frac{\theta(h) - \theta_r}{\theta_s - \theta_r} = (1 + (\alpha h)^n)^{-m} \tag{2}$$

$$K(\theta) = K_s \Theta^\tau [1 - (1 - \Theta^{1/m})^m]^2, \tag{3}$$

where $\theta_s$ and $\theta_r$ denote saturated and residual water contents, respectively, $\alpha$ ($m^{-1}$), $n$ (–) and $m$ ($m = 1 - 1/n$) (–) are empirical parameters shaping the retention curve, $K_s$ is the hydraulic conductivity at saturation condition, and $\tau$ is an empirical parameter shaping the hydraulic function, and is commonly set to 0.5 (Mualem, 1976). With this, there are five unknown parameters $p = \{\alpha, n, \theta_r, \theta_s, K_s\}$ to describe the hydraulic dynamics of the thawed active layer.

**2.2 The parameter estimation**

Given a time series of GPR observations of soil water content in the thawed active layer (Fig. 1c) and weather conditions, the parameters are estimated with the following three steps.

In step 1, the van Genuchten parameters $p_1 = \{\alpha, n, \theta_r, \theta_s\}$ were estimated using the first GPR observation before the rainfall event, as inspired by the estimation of the initial state presented in Jaumann and Roth (2017). Since the subsequent water redistribution leads to a higher water storage in the active layer with a deeper thawing depth, the 2D water storage

distribution is influenced by the thawing structure. Thus, a static hydraulic equilibrium at the initial stage is assumed that a water table at the lower boundary. Correspondingly, the van Genuchten parameters can be derived by fitting the relationship between observed soil water content (storage) and matric potential (depth).

In step 2, the effective soil hydraulic parameters $p_2 = \{\alpha, n, \theta_r, \theta_s, K_s\}$ were determined by minimizing the differences between observed water storages $l_{obs}(x, t)$ and simulated water storages $l_{mod}(x, p, t)$ at location x as an objective function

$$\chi^2(p) = \frac{1}{N} \frac{1}{M} \sum_{t}^{N} \sum_{x}^{M} [l_{obs}(x, t) - l_{mod}(x, p, t)]^2. \tag{4}$$

The Levenberg-Marquardt algorithm as implemented in Jaumann and Roth (2017) is used to minimize $\chi^2(p)$. $M$ is the number of the grid cells in x-dimension ($M = 100$) and $N$ is the number of observations ($N = 7$). Convergence typically requires less than 10 iterations. The optimization procedure also yields correlation coefficients for the resulting parameters. As the gradient-based optimization methods lead to local convergence, the Levenberg-Marquardt algorithm relies on good initial parameters. To address this, we used an ensemble of 50 inversion runs with different initial parameter $K_s$ together with the initial estimated van Genuchten parameters from step 1.

In step 3, the resulting parameters $p_3 = \{\alpha, n, \theta_r, \theta_s, K_s\}$ are taken from the ensemble member with minimal $\chi^2$.

The framework of the 2D inversion procedure is shown in Fig. 2. In step 1, the parameter estimation was solved using the function fminsearch from Matlab (Version: R2015a), which uses the Nelder-Mead-Simplex algorithm. In step 2, the Richards equation solver (muPhi, Ippisch et al., 2006) was used to simulate the spatiotemporal soil water dynamics, and the Levenberg-Marquardt algorithm was used to minimize the differences between the simulated state and observed state. In order to reduce the impact of the inversions with local minima, a mean value of the ensemble inversions with cost function values within 1-sigma of all (34 in 50) is used to assess the effect of the amplitude of the undulating frost table on the parameter estimation. In step 3, the final estimated parameters were determined from the 50 ensemble inversions by selecting the best one with the smallest cost function value. To assess the impact of the white Gaussian noise in GPR observations of soil water storage, the standard errors of the estimated parameters are calculated using 10 ensembles.

**3 Case study**

Here we used synthetic studies to demonstrate the proposed approach and assess the effects of undulating structure on the accuracy of the resulting material properties. The setup of the hydraulic model is shown with Fig. 1. The domain ($12\,\text{m} \times 2\,\text{m}$) comprises a thawed layer and a frozen layer separated by an undulating frost table (Fig. 1b), and have the same sandy soil. The true soil hydraulic parameters of the active layer are listed in Table 1. As the Rechards solver requires a structured rectangular grid, the frozen layer has to be included into the domain. Due to numerical reasons, the according hydraulic conductivity has to be larger than 0. Hence we chose to scale the parameters of the active layer with a Miller scaling factor of 10 ensuring a constant hydraulic conductivity which is 1% of the one of the thawed active layer. The infiltration is represented with a

Neumann flux at the upper boundary and the groundwater table at the lower boundary is represented setting the Dirichlet potential to 0. The boundaries at the sides are impermeable.

The soil water dynamics were simulated with a time step of 100 minutes over a short period of 5.9 days. The time series of the forcing at the upper boundary by rainfall is shown in Fig. 1a. Seven snapshots of soil water storage observations were created using the forward simulations and adding a white Gaussian noise to the evaluated soil water storage associated with an extent of 1m and a mean soil water content of 15%. We assume that the uncertainty of the resulting depth is 0.05 m (Pan et al., 2012a), and the uncertainty of water storage is deduced as 0.15×0.05 = 0.0075 m.

Since the non-uniform change of soil water storage is essential to the inversion, one controlling factor is the undulating structure of frost table. To investigate its influence, three active layers (S1, S2, and S3 in Fig. 3) with different undulating amplitudes (0.25 m, 0.5 m, and 0.75 m, respectively) were investigated. Additionally, the parameter estimations were repeated 10 times using the GPR observations with the same uncertainty but different realizations of the random errors, to investigate the influence of the errors in GPR observations on the approach. The 1500 2D inversions were run on a cluster using parallel computation requiring specifically 192 processors for 10 days.

**4 Results and discussion**

Figure 4 shows the results of the parameter estimation in step 2. On the left panel, the estimated water retention curves of three structures (S1, S2, and S3) can be compared with the synthetic one after step 1. The middle and right panels visualize the ensemble estimates of water retention curve and hydraulic conductivity curve, respectively, together with the synthetic ones after step 2. In each plot, the curves represent the best 34 estimates, according to their $\chi^2$, accounting for 68% (1 σ) of the 50 ensemble inversions. The darker the curve, the better is the estimation. Comparing the best $\chi^2$ for step 1 and the mean values of the 34 best $\chi^2$ for step 2, we found that $\chi^2_{best,S1} > \chi^2_{best,S2} > \chi^2_{best,S3}$. Therefore, the larger undulating amplitude, the better are the estimated hydraulic material properties. We attribute this to the increasing intensity of lateral water redistribution. Generally, the proposed approach works well for the studied cases.

The final estimates of parameters $p_3 = \{\alpha, n, \theta_r, \theta_s, K_s\}$ were derived from the best one, according to $\chi^2$, among the 50 ensemble inversions through step 3. The effects of the white Gaussian noise and the thawing structure on the inversion of the five van Genuchten-Mualem parameters are shown in Fig. 5. The histograms show the estimated parameters from step 3 together with the ones from step 1. Through the 10 ensembles, we can find that the impact of the white Gaussian noise on the parameter estimation is much more significant in step 1 than in step 2. Given an assumption of equilibrium state, the curve fitting is sensitive to the noise due to the narrow cover range of the matric head over the 2D transect. Based on this preliminary estimation, the sensitivity of the parameter estimation in step 2 is much less. In addition, the larger the amplitude of the undulating structure, the smaller are the standard errors. Overall, the results from step 3 show the robustness of the method to the errors of GPR observations, and the mean standard errors for S1, S2 and S3 are 0.40, 0.45, 0.0, 0.02, 8·10⁻⁴ for the five parameters, respectively.

Apart from the white Gaussian noise, the autocorrelation of the parameters is also investigated. The resulting correlation coefficients of the parameters of the best ensemble member for S3 are given in Table 2. The hydraulic material functions are not unique in the sense that different parameter sets produce similar hydraulic material functions within a small intervals of hydraulic head or hydraulic conductivity. As the proposed method merely evaluates the integrated soil water storage, it is less sensitive on specific characteristics of the material functions governed by the individual parameters rather than on the material functions as a whole. Hence, the method yields the inherent parameter correlations of the parameterization model for the material properties. Therefore, the resulting parameters are expected to vary for different ensemble members, the resulting material functions, however, stay within a narrow interval (Fig. 4) indicating high sensitivity of the method on the material properties. Generally, the proposed method works well for the studied case, and the structure of frost table determines the significance of lateral water redistribution. In addition, precipitation features like intensity and duration can strengthen or weaken the effect of the structure on the lateral water redistribution. In essence, parameter estimation of soil water dynamics depends on the coverage of possible hydraulic states by the measurement data, e.g., regulated by precipitation features (e.g., Steenpass et al., 2011; Scharnagl et al., 2011).

For practical application, there are some necessary conditions for this approach to work. First of all, a continuous and undulating frost table is required that leads to lateral water redistribution. Applying this approach for a three-dimensional frost table could be handled in analogy. Secondly, for the studied case, prerequisite conditions like large rainfall intensity and good soil permeability are necessary which are only available at specific regions, e.g., the northeastern QTP. Finally, to capture the lateral water redistribution, selecting the time-slice observations is essential. As a rule of thumb, time-evenly distributed observations are best during the infiltration process. These requirements limit the range of possible applications. However, these limitations can be alleviated when applying this approach at seasonal-scale. Yet, this requires including evapotranspiration and a variable frost table in the model. Since the process of lateral water redistribution does not only rely on precipitation but also melt water by thawing. The latter one is relatively slow, but the amount can be considerable in wet active layers. Here, the timing of GPR observations will not be as sensitive as in the rain-based application.

**5 Summary**

Permafrost tables are often undulating and therefore lead to lateral redistribution of water within the active layer. We propose a method for the inverse estimation of soil hydraulic material properties that exploits this situation. It uses the observed data of thawing depth and soil water storage (which can both be derived from GPR measurements simultaneously) and a 2D simulation of the soil hydrology. We demonstrate this method using synthetic data. Based on a single rain event, seven snapshots of soil water storage were sufficient to accurately estimate the true material properties. The reasonable accuracy of the estimated parameters indicates the good feasibility of the proposed method at desired conditions. The applicability of this method depends on the site of lateral water redistribution, which is mainly controlled by the undulating frost table in the studied case.

As a conceptual study, we assume perfect model and observations with a white Gaussian noise. We comment that GPR works best in coarse-textured soils and may not work at all silty or clay soils. This coincides with the applicability of the hydraulic method, however. Despite its limitations, this approach provides one step forward to accurately estimate hydraulic parameters at the field scale. Its major advantages include non-destructive observations, a fast gain of field-scale soil hydraulic material properties in application.

**Acknowledgements**

We acknowledge the support by the German Research Foundation (DFG) through project RO 1080/12-2, and the National Natural Science Foundation of China through project 41771262, the state of Baden-Württemberg through bwHPC as well as the German Research Foundation (DFG) through grant INST 35/1134-1 FUGG.

**References**

[revised manuscript text omitted]

Table 2 Example of the resulting correlation coefficients for the parameters for one realization of S3 ensemble inversions.

| | $\alpha$ | n | $\theta_r$ | $\theta_s$ | $\log_{10}(K_s)$ |
|---|---|---|---|---|---|
| $\alpha$ | 1 | -0.91 | -0.91 | 0.55 | 0.79 |
| n | -0.91 | 1 | 0.88 | -0.75 | -0.91 |
| $\theta_r$ | -0.91 | 0.88 | 1 | -0.57 | -0.72 |
| $\theta_s$ | 0.55 | -0.75 | -0.57 | 1 | 0.93 |
| $\log_{10}(K_s)$ | 0.79 | -0.91 | -0.72 | 0.93 | 1 |

[Figure]

**Figure 1. An example of the forward modeling. (a) Water flux through the upper boundary. Red circles show the times of seven snapshots of soil water storage observations. (b) The structure of a thawing active layer. (c) A time series of soil water storage observations in the thawed layer, corresponding to the time markers in (a).**

[Figure]

Figure 2. The framework of the 2D inversion procedure.

[Figure]

**Figure 3. The active layer with different thawing structures. The amplitudes of the undulating frost table are (a) S1: 0.25 m; (b) S2: 0.5 m, and (c) S3: 0.75 m, respectively.**

[Figure]

**Figure 4. A comparison of estimated water retention curves (step1 and step2) and hydraulic conductivity curve with the synthetic ones for three structures (S1, S2, and S3). Left panel: initial estimates of water retention curve. Middle panel: final estimates of water retention curve with the best 34 ensemble inversions. Right panel: final estimates of hydraulic conductivity curve with the best 34 ensemble inversions.**

[Figure]

**Figure 5.** Effects of thawing structure on the inversion of the five van Genuchten-Mualem parameters. Given 10 repeated GPR observations with different random errors, histograms of the initial (step 1, light red bars) in and final (step 3, blue bars) estimated parameter values are shown. Only blue bars are shown in the right most panel because there are no initial estimates in step 1 for $K_s$. Dashed lines show the true parameter locations.

---

## Author Comment (AC2) · 12 Sep 2017

**Reply to Referee #2:**

**GENERAL COMMENTS**
*The manuscript describes a synthetic case study, in which a two-dimensional numerical model is used to estimate soil hydraulic parameters in idealized cross sections consisting of a permeable surface layer underlain by an impermeable bottom layer having undulating surface. Texts are generally well organized and written, and modelling approaches are technically sound. However, I am not convinced of the relevance and usefulness of the present manuscript in cryospheric science. It is an interesting numerical exercise, but the manuscript can be made much stronger and interesting to the readership of The Cryosphere. I will list my suggestions in specific comments below*
**Reply:** We thank the reviewer for the constructive comments and suggestions. We revised the manuscript and thus refer to the revised manuscript.

**SPECIFIC COMMENTS**
*1. Title. The study is motivated by GPR applications in the active layer, but the current version of the manuscript presents nothing specific to GPR or the active layer. For example, it does not deal with uncertainties and non-uniqueness in the relationship between dielectric permittivity, which is estimated by GPR, and volumetric water content. The model uses a two-layer structure consisting of permeable and impermeable soils separated by an undulating boundary, which is not specific to the active layer. The permeability of frozen soil is controlled by temperature, but the model does not account for coupled heat-energy transfer processes. Therefore, I think that the current title is somewhat misleading. It will be much better if the authors develop the paper to something that truly describes what is in the title.*
**Reply:**
We don't agree that the manuscript presents nothing specific to GPR or the active layer. First of all, our study is completely based on the context of specific observations of soil depth and water content in 2D and the specific active layer with undulating thawing table, which is not common in warm regions. Given the well-established GPR observations, we decide to not use the deeper GPR information (dielectric permittivity or amplitude of electricmagnetic waves). As a preliminary study, we did not consider the effect of temperature on the permeability of frozen soil in the present model, but we gave an outlook of future development when being used for a seasonal case study.

Given the focus of the study, we changed the title to "Towards the estimation of hydraulic properties of active layers using ground-penetrating radar (GPR) and inverse hydrological modeling"

*2. P3, L7. This form of the Richards equation is incorrect. The gravity term should have a unit vector, not a scalar "1". Also, please define the direction of z-axis.*
**Reply:** We changed the manuscript accordingly.

*3. P3, L22. What value was used for the Dirichlet upper boundary condition? How was it determined?*
**Reply:** We agree. We did use a flux boundary condition. The description of boundary conditions is rephrased in P4 L30 and P5 L1-2 as "The infiltration is represented with a Neumann flux at the upper boundary and the groundwater table at the lower boundary is represented setting the Dirichlet potential to 0. The boundaries at the sides are impermeable."

*4. P3, L25. GPR measures the travel time and amplitude of reflected radar waves, not the amount of soil water storage. The estimation of soil water storage from GPR data is not straight forward and has a large degree of uncertainty. To make this study relevant to The Cryosphere, it is highly desirable to incorporate uncertainties and non-uniqueness in GPR signal interpretation into numerical inversion. I believe that there is an established body of literature on this subject matter.*
**Reply:** This study focuses mainly by GPR observations – soil water content and soil depth (integrated as soil water storage). As summarized in the introduction, the type of multi-channel GPR is an established tool to simultaneously measure soil water content and thawing depth. We choose to stand on this basis other than the step from original GPR signal interpretation and directly use the synthetic GPR observations incorporating real uncertainty in the inverse modeling. Thus, we could focus on the effect of undulating thawing table on the inverse modeling using the standard GPR observations.

*5. P3, L27 - P4, L2. I do not understand this sentence. Please rephrase.*

**Reply:** Changed word "later" to "lateral".

*6. P4, L2-3. It is assumed that the water table (i.e. matric potential = 0) is at the lower boundary. Does the lower boundary refer to the boundary between thawed and frozen soil? If so, does this "static condition" make sense hydrologically? For example, what is the condition at time step 7 in Figure 1? Should that be a more logical representative of the static condition after the complete drainage of the active layer?*
**Reply:** As an initial condition, we assume the water table (i.e. matric potential = 0) is at the lower boundary, which refers to the bottom of the frozen layer. It is generalized as an averaged permafrost table. We acknowledge that this initial assumption might deviate from actual conditions, but it still can constrain the retention curve somehow and provide better starting points of the parameters for step 2.

*7. P4, L5. It appears that a homogeneous soil is used in the model. It is well known that the near surface soil in natural environments is highly heterogeneous both vertically and horizontally. This severely limits the usefulness of the proposed approach to determining soil hydraulic property. I see this as a major weakness of this manuscript. It can be made much stronger by explicitly treating soil heterogeneity in numerical inversion.*
**Reply:** We agree that the near surface soil in natural environments is highly heterogeneous. However, we can cope with it effectively when using the proposed method. Firstly, we could identify evident structural heterogeneity from GPR radargrams. Then we can exclude this case when using the proposed method. Secondly, given a certain error in the GPR observations, our inverse modeling yields effective hydraulic parameters, which certainly represents some small-scale heterogeneity. Thirdly, sandy active layers with weakly heterogeneous top soil are wide spread in alluvial fans in cryosphere regions, which would fit for the studied case.

*8. Figure 1. It appears that a constant flux was applied to the upper model boundary, whereas the method section states that the upper boundary had a Dirichlet condition (P3, L22). What was the actual boundary condition?*
**Reply:** See reply 3.

*9. Table 1. The alpha values should be positive. The pore-size distribution coefficient (n) has a high value, and the residual water content is zero. I would say this is rather an unusual soil. Is this a good representative of typical soils in natural environments? Was this unusual soil purposely chosen for the synthetic case study? Why?*
**Reply:** The product $\alpha \cdot h$ has to be positive, not $\alpha$ itself. In this study, we define $h$ as negative for unsaturated soils, hence $\alpha$ also must be negative.
The parameters were derived from a sandy soil sample in lab experiments, and were selected to demonstrate the approach.

*10. P5, L27-28. As the authors acknowledge, the water distribution over an irregular frost table is inherently three-dimensional. Two-dimensional models provide a useful tool for theoretical discussion, but its utility for practical application is limited. In addition, soil heterogeneity and a high degree of uncertainty in GPR data interpretation makes the present approach impractical to use in active-layer studies in natural environments. I suggest that the authors develop a full-length paper describing the development and application of a more realistic and useful inversion model using actual field examples of GPR data*
**Reply:** We agree that a full-fledged demonstration including experimental verification of the effective parameters gained would be preferable. To the best of our understanding, this is not yet feasible. Still, the concept appears valuable to us and worth a brief communication.

**Brief communication: Toward the estimation of hydraulic properties of active layers using ground-penetrating radar (GPR) and inverse hydrological modeling**

Xicai Pan[1,2], Stefan Jaumann[1,3], Jiabao Zhang[2], and Kurt Roth[1,4]

[1]Institute of Environmental Physics, Heidelberg University, Im Neuenheimer Feld 229, 69120, Heidelberg, Germany
[2]Fengqiu Agro-ecological Experimental Station, State Key Laboratory of Soil and Sustainable Agriculture, Institute of Soil Science, Chinese Academy of Sciences, 71 East Beijing Road, Nanjing, China
[3]HGSMathComp, Heidelberg University, Im Neuenheimer Feld 205, 69120 Heidelberg, Germany
[4]Interdisplinary Center for Scientific Computing (IWR), Heidelberg University, Im Neuenheimer Feld 205, 69120 Heidelberg, Germany

*Correspondence to*: Xicai Pan (xicai.pan@issas.ac.cn)

**Abstract.** Estimation of hydraulic properties of active layers is challenging due to the freeze-thaw effect in space and time. Provided an active layer with an undulating frost table, monitoring of spatial soil water dynamics in the thawed layer could provide significant information for commonly-used inverse estimation of soil hydraulic material properties. In this study, we assess the feasibility to estimate effective soil hydraulic properties using two-dimensional measurements of soil water storage and thawing depth, which can be efficiently derived from ground-penetrating radar (GPR) observations. The results of this study conceptually demonstrate that spatial and temporal observations of soil moisture in the active layer during a rain event are sufficient for inverse estimation of soil hydraulic parameters. The proposed method depends on the lateral water redistribution controlled by the undulating frost table. We suggest that this method could be used for seasonal-scale estimation of soil hydraulic properties.

**1 Introduction**

In permafrost regions, the active layer is a key stratum that controls the exchange of water and energy fluxes between the land surface and the atmosphere. However, the temporal development of the thawing front is hard to monitor at large scales because of its high spatial variability. The variability is not only a result of the local variation in surface features like microtopography and vegetation cover but is also related to subsurface soil properties which govern hydrological processes. Permafrost models such as empirical and process-based ones are normally used to predict the evolution of active layer and permafrost thickness (Riseborough et al., 2008). However, their predictions for thawing depth and permafrost degradation are still barely satisfactory, as this, e.g., lack of high spatial resolution of soil information like hydraulic material properties. Therefore, gaining accurate soil properties is crucial for understanding permafrost degradation and the associated permafrost hydrology. This is particularly important for permafrost studies in the regions with thick and unsaturated active layers like the Qinghai-Tibetan

Plateau (QTP). Soil properties controlling active layer dynamics are the thermal and hydraulic capacities and conductivities. Since thermal properties are less variable than the hydraulic ones, they are mostly derived from soil texture information and empirical models as well as literature data. In contrast, the soil hydraulic material properties are very sensitive to small variation in soil texture or even small-scale structural patterns. Thus, a site-specific determination of hydraulic material properties is
5    essential for permafrost modeling.

Modeling water dynamics in permafrost soils is difficult due to the highly nonlinear hydraulics. Particularly, the soil hydraulic material properties are affected by considerable deformation and transport of soil material during freeze-thaw cycles (Ray et al., 1983; Boike et al., 1998). Additionally, lateral water redistribution is not negligible due to the high variability of the spatial thawing rate (e.g., Quinton et al., 2000; Wright et al., 2009). These features are adverse for the one-dimensional
10    (1D) model, but this provides various possibilities for two-dimensional (2D) inverse modeling if spatial observations of the thawing front and the soil moisture are available. These spatial observations contain significant information as they monitor soil water dynamics. Lateral water redistribution is common in thawing active layers with undulating frost tables, and happens continuously during the thawing season. This rapid lateral water redistribution also occurs after strong rainfall events, which is often seen in the permafrost regions on the QTP (Pan et al., 2014), where precipitation is dominated by summer monsoon.
15    Inverse methods are commonly used to estimate hydraulic material properties from observed state variables at different scales. Progress in inverse modeling of soil hydraulic properties has been reviewed by Vrugt et al. (2008). Generally, results from the inverse method using in-situ 1D monitoring profile are limited to point scale, and this approach is usually hard to be used larger scales. Conversely, recent developments of soil hydraulic property inversion using satellite remote sensing yield some useful results at larger scales, e.g., from field-scale to catchment scale, which, however, are mostly based on the shallow
20    measurement depth (< 5 cm) and an assumption of homogeneous soil column (Mohanty, 2013). In fact, root zone soils are heterogeneous whereas observations of the top soils are not representative. Accordingly, the model predictions using estimated hydraulic parameters solely based on near-surface soil moisture observations are not as good as the ones based the observations for the entire profile (Bandara et al., 2014). Furthermore, to yield robust results, all inverse parameter estimation methods rely on a time series of measurements that must contain significant information on soil water dynamics (Bandara et al., 2013).
25    Geophysical methods like ground-penetrating radar (GPR) of detecting soil layer boundaries developed rapidly (e.g., Neal, 2004; Annan, 2005; Jol and Bristow, 2003) and was extended for in real-time imaging of near-surface/layering soil water content (e.g., Huisman et al., 2003; Weihermüller, et al., 2007; Bradford, 2008; Pan et al., 2012b; Klenk et al., 2016). For permafrost soils, thawing depth of active layer and soil water content can be also simultaneously retrieved (e.g., Gerhards et al., 2008, Westermann et al., 2010), hence the total soil water storage can be derived accordingly as an integrated value of
30    both. Quantitative observations of spatiotemporal variation in thawing depth and soil water content within plot-scale soil were successfully demonstrated on the QTP by Wollschläger et al. (2010) and Pan et al. (2014). As a primary modeling step, we go for a representation of the possibly heterogeneous soil as a set of uniform layers. The object then is to determine the effective hydraulic material properties, which we here propose to do based on GPR measurements. Whether such an effective representation is reasonable or not can be judged from the measured radargram.

Undulating frost tables are common in the permafrost regions resulting from the patterned surfaces such as vegetation cover, snow cover, and soil properties. Provided significant lateral water redistribution induced by an undulating frost table in active layers and spatiotemporal GPR observations, we investigate the applicability of the inverse modeling approach to efficiently estimate effective hydraulic material properties. In this study, we use synthetic data and discuss the effect of the amplitude of the undulating frost table on the estimating effective soil hydraulic material properties.

**2 Scheme of hydraulic parameter estimation**

**2.1 The 2D hydrological model**

Generally, two-dimensional Darcian water flow in a variably-saturated isotropic medium is described with Richards equation (Richards, 1931)

$$\partial_t \theta(h) - \nabla \cdot [K_w(\theta(h))[\nabla h - e_z]] = 0, \tag{1}$$

where $\theta$ is the volumetric soil water content, ($m^3 m^{-3}$), $K_w$ is the hydraulic conductivity, ($ms^{-1}$), and $h$ is the matric head, (m), unit vector in z-direction $e_z$ indicating the direction of gravity. The material functions involving soil hydraulic parameters in the Richards equation compose the hydraulic conductivity function $K_w$ (h) and the soil water characteristic $\theta$(h), usually in terms of the water saturation $\Theta$ (–). A widely employed model for these two relationships is the van Genuchten-Mualem model (van Genuchten, 1980; Mualem, 1976):

$$\Theta(h) = \frac{\theta(h) - \theta_r}{\theta_s - \theta_r} = (1 + (\alpha h)^n)^{-m} \tag{2}$$

$$K(\theta) = K_s \Theta^\tau [1 - (1 - \Theta^{1/m})^m]^2, \tag{3}$$

where $\theta_s$ and $\theta_r$ denote saturated and residual water contents, respectively, $\alpha$ ($m^{-1}$), $n$ (–) and $m$ ($m = 1-1/n$) (–) are empirical parameters shaping the retention curve, $K_s$ is the hydraulic conductivity at saturation condition, and $\tau$ is an empirical parameter shaping the hydraulic function, and is commonly set to 0.5 (Mualem, 1976). With this, there are five unknown parameters $p = \{\alpha, n, \theta_r, \theta_s, K_s\}$ to describe the hydraulic dynamics of the thawed active layer.

**2.2 The parameter estimation**

Given a time series of GPR observations of soil water content in the thawed active layer (Fig. 1c) and weather conditions, the parameters are estimated with the following three steps.

In step 1, the van Genuchten parameters $p_1 = \{\alpha, n, \theta_r, \theta_s\}$ were estimated using the first GPR observation before the rainfall event, as inspired by the estimation of the initial state presented in Jaumann and Roth (2017). Since the subsequent water redistribution leads to a higher water storage in the active layer with a deeper thawing depth, the 2D water storage

distribution is influenced by the thawing structure. Thus, a static hydraulic equilibrium at the initial stage is assumed that a water table at the lower boundary. Correspondingly, the van Genuchten parameters can be derived by fitting the relationship between observed soil water content (storage) and matric potential (depth).

In step 2, the effective soil hydraulic parameters $p_2 = \left\{ \alpha, \text{n}, \theta_r, \theta_s, K_s \right\}$ were determined by minimizing the differences between observed water storages $l_{obs}(x, t)$ and simulated water storages $l_{mod}(x, p, t)$ at location x as an objective function

$$\chi^2(p) = \frac{1}{N} \frac{1}{M} \sum_{t}^{N} \sum_{x}^{M} \left[ l_{obs}(x, t) - l_{mod}(x, p, t) \right]^2. \tag{4}$$

The Levenberg-Marquardt algorithm as implemented in Jaumann and Roth (2017) is used to minimize $\chi^2(p)$. $M$ is the number of the grid cells in x-dimension ($M = 100$) and $N$ is the number of observations ($N = 7$). Convergence typically requires less than 10 iterations. The optimization procedure also yields correlation coefficients for the resulting parameters. As the gradient-based optimization methods lead to local convergence, the Levenberg-Marquardt algorithm relies on good initial parameters. To address this, we used an ensemble of 50 inversion runs with different initial parameter $K_s$ together with the initial estimated van Genuchten parameters from step 1.

In step 3, the resulting parameters $p_3 = \left\{ \alpha, \text{n}, \theta_r, \theta_s, K_s \right\}$ are taken from the ensemble member with minimal $\chi^2$.

The framework of the 2D inversion procedure is shown in Fig. 2. In step 1, the parameter estimation was solved using the function fminsearch from Matlab (Version: R2015a), which uses the Nelder-Mead-Simplex algorithm. In step 2, the Richards equation solver (muPhi, Ippisch et al., 2006) was used to simulate the spatiotemporal soil water dynamics, and the Levenberg-Marquardt algorithm was used to minimize the differences between the simulated state and observed state. In order to reduce the impact of the inversions with local minima, a mean value of the ensemble inversions with cost function values within 1-sigma of all (34 in 50) is used to assess the effect of the amplitude of the undulating frost table on the parameter estimation. In step 3, the final estimated parameters were determined from the 50 ensemble inversions by selecting the best one with the smallest cost function value. To assess the impact of the white Gaussian noise in GPR observations of soil water storage, the standard errors of the estimated parameters are calculated using 10 ensembles.

**3 Case study**

Here we used synthetic studies to demonstrate the proposed approach and assess the effects of undulating structure on the accuracy of the resulting material properties. The setup of the hydraulic model is shown with Fig. 1. The domain (12 m × 2 m) comprises a thawed layer and a frozen layer separated by an undulating frost table (Fig. 1b), and have the same sandy soil. The true soil hydraulic parameters of the active layer are listed in Table 1. As the Rechards solver requires a structured rectangular grid, the frozen layer has to be included into the domain. Due to numerical reasons, the according hydraulic conductivity has to be larger than 0. Hence we chose to scale the parameters of the active layer with a Miller scaling factor of 10 ensuring a constant hydraulic conductivity which is 1% of the one of the thawed active layer. The infiltration is represented with a

Neumann flux at the upper boundary and the groundwater table at the lower boundary is represented setting the Dirichlet potential to 0. The boundaries at the sides are impermeable.

The soil water dynamics were simulated with a time step of 100 minutes over a short period of 5.9 days. The time series of the forcing at the upper boundary by rainfall is shown in Fig. 1a. Seven snapshots of soil water storage observations were created using the forward simulations and adding a white Gaussian noise to the evaluated soil water storage associated with an extent of 1m and a mean soil water content of 15%. We assume that the uncertainty of the resulting depth is 0.05 m (Pan et al., 2012a), and the uncertainty of water storage is deduced as 0.15×0.05 = 0.0075 m.

Since the non-uniform change of soil water storage is essential to the inversion, one controlling factor is the undulating structure of frost table. To investigate its influence, three active layers (S1, S2, and S3 in Fig. 3) with different undulating amplitudes (0.25 m, 0.5 m, and 0.75 m, respectively) were investigated. Additionally, the parameter estimations were repeated 10 times using the GPR observations with the same uncertainty but different realizations of the random errors, to investigate the influence of the errors in GPR observations on the approach. The 1500 2D inversions were run on a cluster using parallel computation requiring specifically 192 processors for 10 days.

**4 Results and discussion**

Figure 4 shows the results of the parameter estimation in step 2. On the left panel, the estimated water retention curves of three structures (S1, S2, and S3) can be compared with the synthetic one after step 1. The middle and right panels visualize the ensemble estimates of water retention curve and hydraulic conductivity curve, respectively, together with the synthetic ones after step 2. In each plot, the curves represent the best 34 estimates, according to their $\chi^2$, accounting for 68% (1 σ) of the 50 ensemble inversions. The darker the curve, the better is the estimation. Comparing the best $\chi^2$ for step 1 and the mean values of the 34 best $\chi^2$ for step 2, we found that $\chi^2_{best,S1} > \chi^2_{best,S2} > \chi^2_{best,S3}$. Therefore, the larger undulating amplitude, the better are the estimated hydraulic material properties. We attribute this to the increasing intensity of lateral water redistribution. Generally, the proposed approach works well for the studied cases.

The final estimates of parameters $p_3 = \{\alpha, \mathrm{n}, \theta_r, \theta_s, K_s\}$ were derived from the best one, according to $\chi^2$, among the 50 ensemble inversions through step 3. The effects of the white Gaussian noise and the thawing structure on the inversion of the five van Genuchten-Mualem parameters are shown in Fig. 5. The histograms show the estimated parameters from step 3 together with the ones from step 1. Through the 10 ensembles, we can find that the impact of the white Gaussian noise on the parameter estimation is much more significant in step 1 than in step 2. Given an assumption of equilibrium state, the curve fitting is sensitive to the noise due to the narrow cover range of the matric head over the 2D transect. Based on this preliminary estimation, the sensitivity of the parameter estimation in step 2 is much less. In addition, the larger the amplitude of the undulating structure, the smaller are the standard errors. Overall, the results from step 3 show the robustness of the method to the errors of GPR observations, and the mean standard errors for S1, S2 and S3 are 0.40, 0.45, 0.0, 0.02, 8·10⁻⁴ for the five parameters, respectively.

Apart from the white Gaussian noise, the autocorrelation of the parameters is also investigated. The resulting correlation coefficients of the parameters of the best ensemble member for S3 are given in Table 2. The hydraulic material functions are not unique in the sense that different parameter sets produce similar hydraulic material functions within a small intervals of hydraulic head or hydraulic conductivity. As the proposed method merely evaluates the integrated soil water storage, it is less sensitive on specific characteristics of the material functions governed by the individual parameters rather than on the material functions as a whole. Hence, the method yields the inherent parameter correlations of the parameterization model for the material properties. Therefore, the resulting parameters are expected to vary for different ensemble members, the resulting material functions, however, stay within a narrow interval (Fig. 4) indicating high sensitivity of the method on the material properties. Generally, the proposed method works well for the studied case, and the structure of frost table determines the significance of lateral water redistribution. In addition, precipitation features like intensity and duration can strengthen or weaken the effect of the structure on the lateral water redistribution. In essence, parameter estimation of soil water dynamics depends on the coverage of possible hydraulic states by the measurement data, e.g., regulated by precipitation features (e.g., Steenpass et al., 2011; Scharnagl et al., 2011).

For practical application, there are some necessary conditions for this approach to work. First of all, a continuous and undulating frost table is required that leads to lateral water redistribution. Applying this approach for a three-dimensional frost table could be handled in analogy. Secondly, for the studied case, prerequisite conditions like large rainfall intensity and good soil permeability are necessary which are only available at specific regions, e.g., the northeastern QTP. Finally, to capture the lateral water redistribution, selecting the time-slice observations is essential. As a rule of thumb, time-evenly distributed observations are best during the infiltration process. These requirements limit the range of possible applications. However, these limitations can be alleviated when applying this approach at seasonal-scale. Yet, this requires including evapotranspiration and a variable frost table in the model. Since the process of lateral water redistribution does not only rely on precipitation but also melt water by thawing. The latter one is relatively slow, but the amount can be considerable in wet active layers. Here, the timing of GPR observations will not be as sensitive as in the rain-based application.

**5 Summary**

Permafrost tables are often undulating and therefore lead to lateral redistribution of water within the active layer. We propose a method for the inverse estimation of soil hydraulic material properties that exploits this situation. It uses the observed data of thawing depth and soil water storage (which can both be derived from GPR measurements simultaneously) and a 2D simulation of the soil hydrology. We demonstrate this method using synthetic data. Based on a single rain event, seven snapshots of soil water storage were sufficient to accurately estimate the true material properties. The reasonable accuracy of the estimated parameters indicates the good feasibility of the proposed method at desired conditions. The applicability of this method depends on the site of lateral water redistribution, which is mainly controlled by the undulating frost table in the studied case.

As a conceptual study, we assume perfect model and observations with a white Gaussian noise. We comment that GPR works best in coarse-textured soils and may not work at all silty or clay soils. This coincides with the applicability of the hydraulic method, however. Despite its limitations, this approach provides one step forward to accurately estimate hydraulic parameters at the field scale. Its major advantages include non-destructive observations, a fast gain of field-scale soil hydraulic

5   material properties in application.

**Acknowledgements**

We acknowledge the support by the German Research Foundation (DFG) through project RO 1080/12-2, and the National Natural Science Foundation of China through project 41771262, the state of Baden-Württemberg through bwHPC as well as the German Research Foundation (DFG) through grant INST 35/1134-1 FUGG.

10  **References**

[revised manuscript text omitted]

Table 2 Example of the resulting correlation coefficients for the parameters for one realization of S3 ensemble inversions.

|  | $\alpha$ | n | $\theta_r$ | $\theta_s$ | $\log_{10}(K_s)$ |
|---|---|---|---|---|---|
| $\alpha$ | 1 | -0.91 | -0.91 | 0.55 | 0.79 |
| n | -0.91 | 1 | 0.88 | -0.75 | -0.91 |
| $\theta_r$ | -0.91 | 0.88 | 1 | -0.57 | -0.72 |
| $\theta_s$ | 0.55 | -0.75 | -0.57 | 1 | 0.93 |
| $\log_{10}(K_s)$ | 0.79 | -0.91 | -0.72 | 0.93 | 1 |

[Figure]

**Figure 1. An example of the forward modeling. (a) Water flux through the upper boundary. Red circles show the times of seven snapshots of soil water storage observations. (b) The structure of a thawing active layer. (c) A time series of soil water storage observations in the thawed layer, corresponding to the time markers in (a).**

[Figure]

**Figure 2. The framework of the 2D inversion procedure.**

[Figure]

**Figure 3. The active layer with different thawing structures. The amplitudes of the undulating frost table are (a) S1: 0.25 m; (b) S2: 0.5 m, and (c) S3: 0.75 m, respectively.**

[Figure]

**Figure 4. A comparison of estimated water retention curves (step1 and step2) and hydraulic conductivity curve with the synthetic ones for three structures (S1, S2, and S3). Left panel: initial estimates of water retention curve. Middle panel: final estimates of water retention curve with the best 34 ensemble inversions. Right panel: final estimates of hydraulic conductivity curve with the best 34 ensemble inversions.**

[Figure]

**Figure 5. Effects of thawing structure on the inversion of the five van Genuchten-Mualem parameters. Given 10 repeated GPR observations with different random errors, histograms of the initial (step 1, light red bars) in and final (step 3, blue bars) estimated parameter values are shown. Only blue bars are shown in the right most panel because there are no initial estimates in step 1 for $K_s$. Dashed lines show the true parameter locations.**